# Reconciling persistent and dynamic hypotheses of working memory coding in prefrontal cortex

Sean E. Cavanagh [1], John P. Towers[1], Joni D. Wallis[2,3], Laurence T. Hunt [1,4,5] & Steven W. Kennerley [1,2,3]

Competing accounts propose that working memory (WM) is subserved either by persistent activity in single neurons or by dynamic (time-varying) activity across a neural population. Here, we compare these hypotheses across four regions of prefrontal cortex (PFC) in an oculomotor-delayed-response task, where an intervening cue indicated the reward available for a correct saccade. WM representations were strongest in ventrolateral PFC neurons with higher intrinsic temporal stability (time-constant). At the population-level, although a stable mnemonic state was reached during the delay, this tuning geometry was reversed relative to cue-period selectivity, and was disrupted by the reward cue. Single-neuron analysis revealed many neurons switched to coding reward, rather than maintaining task-relevant spatial selectivity until saccade. These results imply WM is fulfilled by dynamic, population-level activity within high time-constant neurons. Rather than persistent activity supporting stable mnemonic representations that bridge subsequent salient stimuli, PFC neurons may stabilise a dynamic population-level process supporting WM.

[1] Sobell Department of Motor Neuroscience, University College London, London WC1N 3BG, UK. [2] Department of Psychology, University of California at Berkeley, Berkeley, CA 94720, United States. [3] Helen Wills Neuroscience Institute, University of California at Berkeley, Berkeley, CA 94720, United States. [4] Max Planck-UCL Centre for Computational Psychiatry and Aging, University College London, London WC1B 5EH, UK. [5] Wellcome Centre for Integrative Neuroimaging, Department of Psychiatry, University of Oxford, Oxford OX37JX, UK. Correspondence and requests for materials should be addressed to S.E.C. (email: sean.cavanagh.12@ucl.ac.uk) or to S.W.K. (email: s.kennerley@ucl.ac.uk)

Temporary maintenance of relevant information in the absence of direct sensory input is a crucial component of working memory (WM). The neuronal basis of WM has been studied extensively through single-neuron recordings. These typically involve subjects performing tasks where a transient sensory stimulus must be remembered across a multi-second delay, before a probe cues a response to the remembered stimulus. A consensus has developed from these experiments[1–3], and from lesion studies[4,5], that cognitive operations that use information in WM depend upon the prefrontal cortex (PFC)[6], with PFC neurons sustaining stimulus-specific representations across mnemonic delays. This stable coding has inspired biophysically plausible attractor models of WM, in which persistent activity is facilitated by a neocortical circuit structured with strong recurrent connections between similarly tuned neurons[7].

Recent findings have challenged these established views. PFC neuronal responses are often highly heterogeneous, with a minority exhibiting prolonged stimulus-specific encoding during delays[8–11]. The majority of neurons instead show short-lived selectivity, with variable onset latencies and durations. This pattern of WM activity is termed dynamic coding. Evidence for dynamic coding has led to revised attractor models that reconcile time-varying and stable single-neuron responses[12]. It has also inspired alternate explanations for how WM may be achieved without relying upon a stable representation through persistent spiking activity[8,13–17]. These include dynamic trajectory models where neural firing preserves a representation of the mnemonic stimulus throughout a delay by moving through a reproducible path of activity[14,16,17]. They also include synaptic models where WM is achieved by short-term plasticity of synaptic weights[8,13]. In the latter, stable delay period WM correlates still arise, but as a by-product of spontaneous activity within a circuit that is temporarily embedded with mnemonic information.

An important prediction rarely tested in the context of WM models relates to how network representations of stimuli resist distraction[18–20]. In a world where we constantly experience salient sensory stimuli, cognitive operations dependent on WM require that mnemonic information is resistant to the processing of other stimuli in our environment. The majority of tasks used to study single-neuron WM correlates lack intervening stimuli during delays. If memoranda are maintained purely by persistent single-neuron activity, and if those neurons flexibly encode multiple task features (as is common in PFC[21–24]), a subsequent salient stimulus could disrupt the attractor state and cause the memory to be distorted or lost. Several neurophysiological accounts suggest PFC possesses a privileged position in cortical processing—the ability for individual task-selective neurons to resist distraction[25–27]. More recently, however, the view that PFC neurons are resistant to distractors has been challenged[20,28]. If WM is maintained in the absence of stable single-neuron representations, it becomes important to understand how memoranda are encoded across the PFC population when subsequent behaviourally relevant salient stimuli are presented, and what role neurons with persistent activity play in such population-level encoding.

Another important consideration is that single neurons exhibit large heterogeneity in the degree to which they exhibit persistent activity at rest[29,30]. By fitting an exponential decay to the autocorrelation of neuronal firing outside of the task, it is possible to characterise individual neurons' intrinsic temporal stability (time-constant)[30,31]. A neuron's time-constant likely reflects a combination of its intrinsic physical properties and its degree of recurrent connectivity[32]. Because neurons with higher time-constants are more likely to maintain information during extended cognitive processes such as decision-making[30], we hypothesised that heterogeneity in single-neuron time-constants may explain why some neurons retain stimulus-specific mnemonic representations across delays, whereas others exhibit more transient selectivity. This would reconcile persistent and dynamic WM coding. If attractor states underlie WM, classical stable mnemonic representations should primarily be evident in a subpopulation of neurons with high time-constants. Furthermore, neurons with high time-constants may facilitate the stability of WM representations throughout distraction.

We tested these hypotheses in an oculomotor-delayed-response task, where a stimulus revealing the reward for a correct response was presented either before or after the spatial cue which had to be maintained in WM. Presentation after the mnemonic cue serves as a salient behaviourally relevant cue, potentially disrupting spatial WM representations[33,34]. This also allowed us to test how an interfering stimulus affected network-level mnemonic coding as a function of neuronal time-constant.

## Results

**Task and neurophysiological recordings**. Two rhesus macaques (*Macaca mulatta*) performed an oculomotor-delayed-response task requiring spatial WM, where the reward amount for successful responses varied across trials (Fig. 1a)[33,34]. Briefly (Methods), subjects were first required to fixate a central cue for 1000 ms. If fixation was maintained, two cues were sequentially presented (for 500 ms apiece), each followed by a 1000 ms delay. The spatial cue indicated which of 24 locations the subject had to hold in WM (the mnemonic stimulus); the reward cue indicated which of 5 reward magnitudes the subject would receive for a saccade to the remembered location. The subject could elicit a saccade to the remembered location following a go cue. In Reward-Space (RS)-trials, the first and second cues were the reward and spatial cues respectively; the cue order was reversed in Space-Reward (SR)-trials.

Single-neuron recordings were taken from four regions of PFC (Fig. 1b, Methods): anterior cingulate cortex (ACC, $n = 198$), orbitofrontal cortex (OFC, $n = 152$), dorsolateral PFC (DLPFC, $n = 205$) and ventrolateral PFC (VLPFC, $n = 139$). All neurons per region were pooled across sessions in order to examine population-level activity[11,12,35].

**Resting time-constants**. We first sought to define each neuron's resting time-constant (tau) by fitting an exponential decay to its spike-count autocorrelation during the 1000 ms fixation period[30]. The autocorrelation functions of those neurons that could successfully be described by an exponential decay with an offset[31] were fitted to yield a resting time-constant for each neuron (358 of 694 single neurons, Methods).

As previously reported[30], there was marked heterogeneity in the temporal specialisation of individual neurons both within and between PFC regions (Fig. 1c). We next characterised the population-level taus of the four PFC brain regions (Fig. 1d, Methods). For this analysis, data from all recorded neurons within each brain area was fitted using the same exponential decay equation. Our results corroborate previous findings, emphasising the ACC at the summit of a time-constant hierarchy across PFC regions[30,31].

**Decoding analysis of WM activity**. We next applied a decoding approach to investigate population-dynamics across PFC[20,35]. Briefly, we trained separate classifiers on population responses to distinguish between each condition (eight collapsed locations for Space; five Reward levels; Methods).

Our results provide the most complete comparison to date of population-level WM activity across PFC regions (Fig. 2). Of the four PFC regions examined, VLPFC activity best discriminated

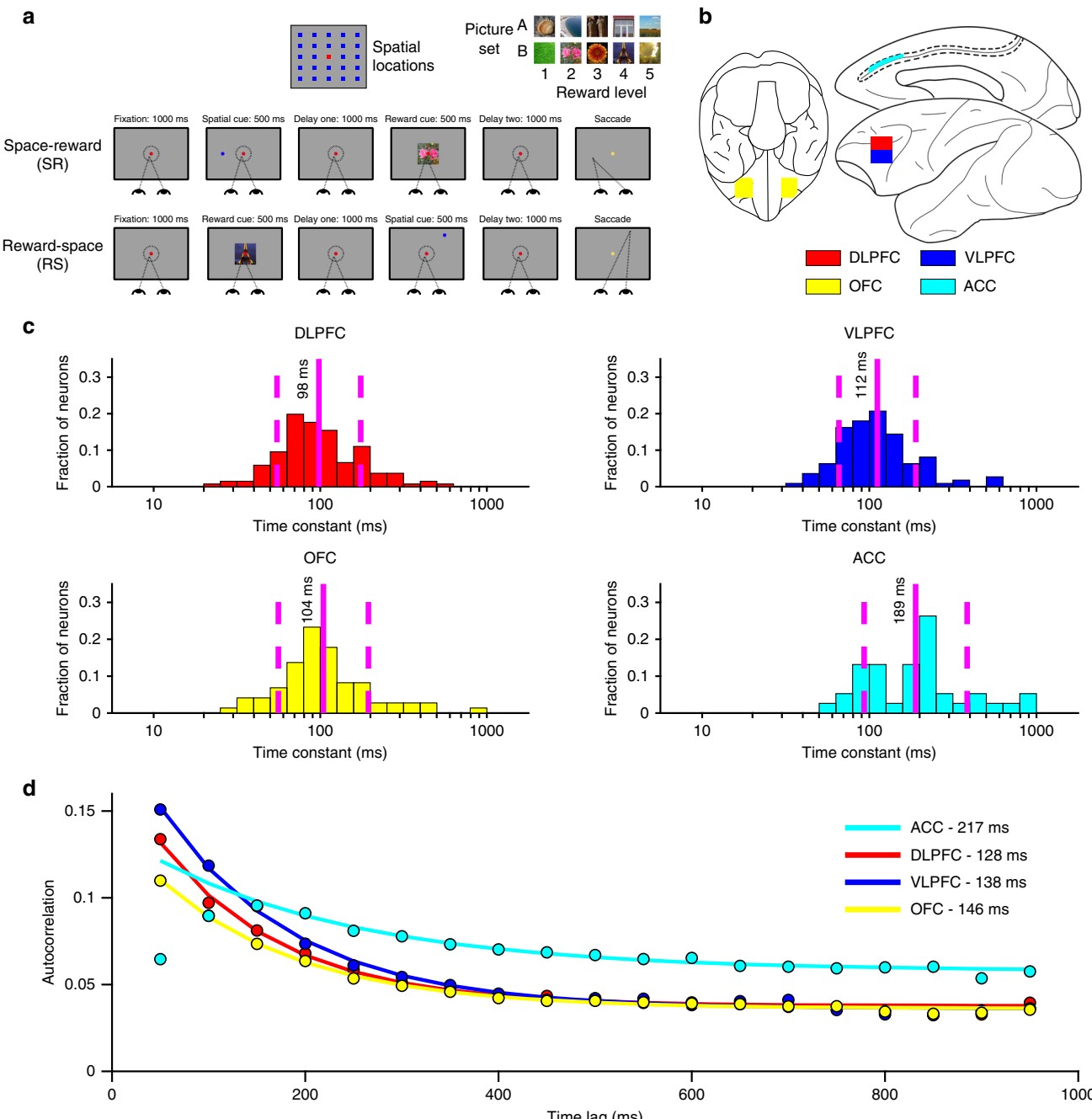

**Fig. 1** Overview of reward-varying oculomotor-delayed-response task, recording locations and time-constant analysis. **a** Reward-varying oculomotor-delayed-response task. Monkeys were trained to remember a spatial position in working memory. They were also presented with a cue indicating the reward size they would receive for successfully completing the trial with a saccade to the remembered location. On SR (Space-Reward) trials, the spatial cue was presented first; whereas on the RS (Reward-Space) trials, the cues were presented in the reverse order. On SR-trials the reward cue, therefore, acted as an intervening behaviourally relevant cue whilst subjects maintained the task-relevant spatial information in working memory. The reward cue images shown are similar, but not identical to those used in the study. **b** Approximate location of neural recordings drawn onto diagrams based upon the Paxinos brain atlas[68] in F99 space[71] viewed using the scalable brain atlas[72]. Neurons were recorded from anterior cingulate cortex (ACC), dorsolateral prefrontal cortex (DLPFC), ventrolateral prefrontal cortex (VLPFC) and orbitofrontal cortex (OFC). **c** Histograms of the single-neuron time-constants within the four PFC brain regions. Time-constants are highly variable across neurons. Time-constants differed significantly across areas (Kruskal–Wallis test, $p = 2.94 \times 10^{-6}$), where the longest taus were within ACC (Mann–Whitney $U$ tests; ACC vs. DLPFC, $p = 5.13 \times 10^{-7}$; ACC vs. OFC, $p = 2.48 \times 10^{-5}$; ACC vs. VLPFC, $p = 2.72 \times 10^{-5}$). Solid and dashed vertical lines represent mean ($\text{Log}(\tau)$) and mean ($\text{Log}(\tau)$) ± SD ($\text{Log}(\tau)$), respectively. **d** Population-level time-constants of firing rate autocorrelation in DLPFC, VLPFC, OFC and ACC during pre-stimulus fixation epoch. Time-constant captures the rate of decay of autocorrelation over time. ACC had the highest and most distinct time-constant of all PFC regions studied

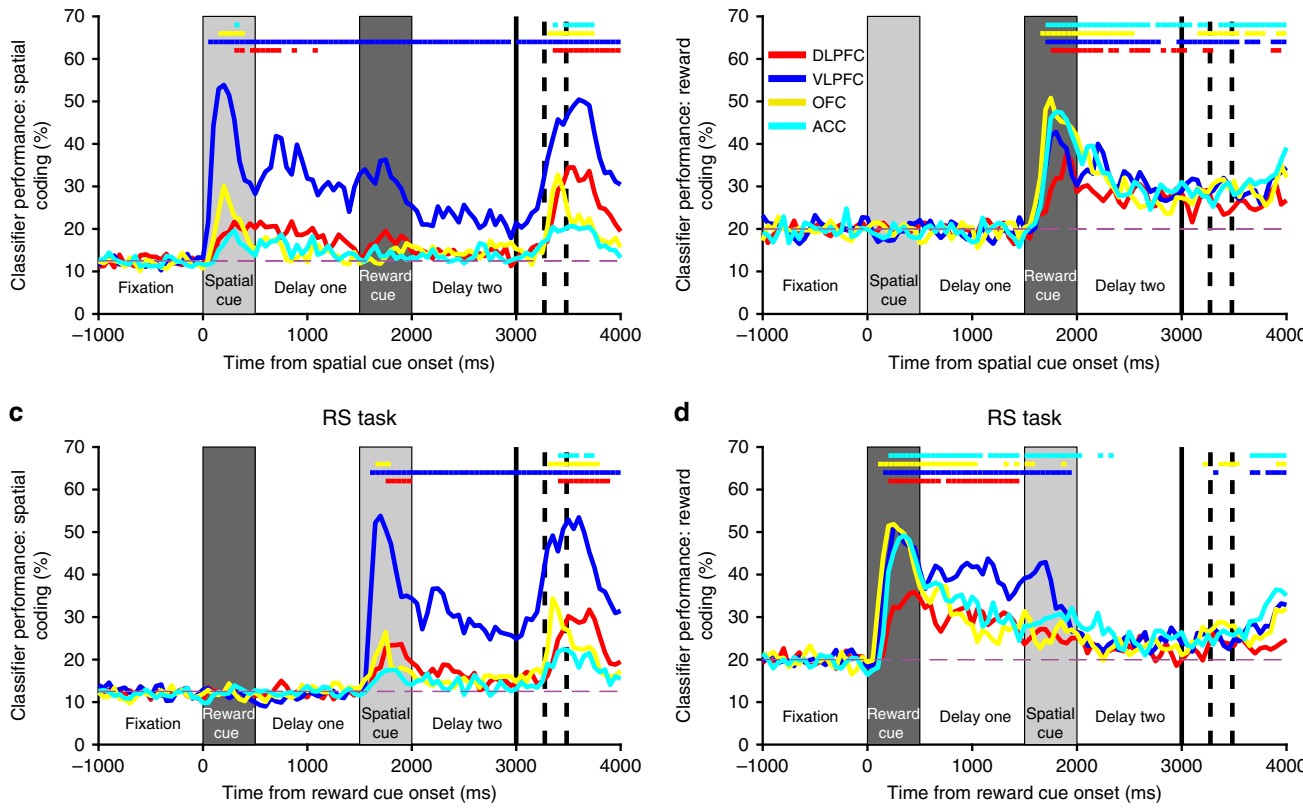

**Fig. 2** Ventrolateral prefrontal neurons maintain information for both spatial and reward stimuli during delay epochs. The mean performance of classifiers (1000 permutations, Methods) trained to decode each task feature (spatial location, **a**, **c**; reward level, **b**, **d**) are plotted for each brain area and trial type (SR-task, **a**, **b**; RS-task, **c**, **d**). Ventrolateral prefrontal cortex (VLPFC) is the only region to strongly code information about space and reward across the trial. Notably, the VLPFC activity primarily encodes the factor most recently presented. When the reward cue is shown first (RS-task, **c**, **d**), a representation of reward size is maintained throughout delay-one, but falls away when the spatial cue is presented. More surprisingly, a similar weakening of spatial coding is also observed on the SR-task (**a**), even though this analysis is restricted to trials where the subject remembered the correct spatial location. The VLPFC population strongly encodes and maintains a representation of the remembered spatial location, but this is substantially weakened by the offset of the reward cue. The first solid vertical line signifies when subjects were cued to respond. The first and second dashed vertical lines represent the average timing of the subjects' saccade and the onset of reward, respectively. Solid coloured horizontal lines represent significant encoding for the corresponding brain region (2.5th percentile of distribution > chance level, $p < 0.05$, Methods). The dashed magenta line represents chance level classifier performance

between both the different spatial locations and the different reward sizes regardless of trial type (SR and RS), and was the only region that sustained both of these selectivity patterns across delays. VLPFC was also the region most strongly discriminating spatial information immediately prior to saccade. An error-trial analysis confirmed VLPFC activity was behaviourally relevant; decoding accuracy was significantly stronger on correct SR-trials relative to error SR-trials. This effect was evident during cue presentation, delay-one, and during saccade preparation (Supplementary Figure 1).

VLPFC activity exhibited a distinctive temporal profile. On SR-trials, spatial information was strongly represented during both the spatial cue epoch and delay-one (Fig. 2a). However, shortly after the reward cue was presented, the VLPFC spatial coding was significantly reduced (Fig. 2a; Supplementary Figure 2C–E). This decline in stability was concomitant with the presentation of the reward cue (Supplementary Figure 2F, G). At the expense of spatial selectivity, a robust representation of reward emerged which was maintained through the end of the trial (Fig. 2b). This reward coding was noteworthy, as retaining a memory of spatial location is the only task variable necessary for correct performance. A similar pattern of selectivity switching was

present in RS-trials, where the VLPFC population initially maintained a representation of the expected reward, but this representation attenuated as the spatial representation strongly emerged following the spatial cue (Fig. 2c, d). These results are consistent with VLPFC spiking-activity prioritising a representation of the most recently attended information, regardless of whether it is necessary to store in WM for successful performance[36,37]. However, while the spatial coding is weakened by reward cue presentation on SR-trials (Fig. 2a), reward coding is reduced to insignificance by spatial cue presentation on RS-trials (Fig. 2d). This suggests that following the presentation of a subsequent stimulus, VLPFC maintains a residual level of coding for mnemonic, but not inessential, information.

Maintenance of spatial discriminability in DLPFC was comparatively weaker, emerging relatively late in the spatial cue epoch and decaying shortly after delay-one (Fig. 2a, c). Further analysis revealed that our results were not dependent upon whether the border between VLPFC and DLPFC was within or just ventral to the principal sulcus (PS) (Supplementary Figure 3). OFC had phasic representations of spatial location during cue presentation and response[38] (Fig. 2a, c), while ACC only exhibited brief spatial selectivity at the time of reward. All

regions had strong coding of reward size in both trial types (Fig. 2b, d).

**Population activity separated by resting time-constant**. We next sought to link the two previous analyses, exploring whether the heterogeneity of single-neuron time-constants (Fig. 1c) predicted different functional roles during WM. As cells with higher time-constants have an intrinsic capacity for sustained persistent activity, we hypothesised that these cells would more likely be integral to stable attractor states and therefore exhibit stronger and more prolonged maintenance of spatial information[7,12]. We focussed upon VLPFC, as this was the only candidate region with sustained spatial selectivity. We subdivided the population based upon a median split of time-constant[30], and then re-computed the spatial and reward classifiers as in Fig. 2 for high (high-tau) and low time-constant (low-tau) subpopulations (Fig. 3).

As hypothesised, the high-tau VLPFC neuronal subpopulation had more sustained selectivity for both spatial and reward information. Both subpopulations showed a similar temporal profile to the whole VLPFC population (Fig. 2), but selectivity in the low-tau neurons decayed quickly following stimulus offset. A formal comparison between the subpopulations indicated the high-tau cells had stronger spatial selectivity from delay-one

onwards in SR-trials and delay-two onwards in RS-trials (Fig. 3a, c). However, an examination of activity when the spatial cue was onscreen revealed strong selectivity that was statistically indistinguishable between the two subpopulations (spatial cue of Fig. 3a, c). In other words, it is not the case that low-tau subpopulations are simply less task-selective. Instead, high-tau cells appear to be specialised for exhibiting sustained selectivity across delays, a property which may be critical for supporting WM processes.

**Cross-temporal activity separated by resting time-constant**. The results presented so far—sustained population-level selectivity across delays only in cells with persistent resting activity—could be explained by both attractor models and alternate hypotheses of WM[7,14]. They are also consistent with previous reports relating baseline autocorrelation to WM activity in single neurons[39]. The population WM selectivity in Fig. 3 could be supported either by individual neurons maintaining strong selectivity across the trial, or neurons dynamically encoding information with different latencies and durations such that the population-level selectivity is maintained over time.

To contrast between these hypotheses, we performed a cross-temporal pattern analysis to probe the stability of the active

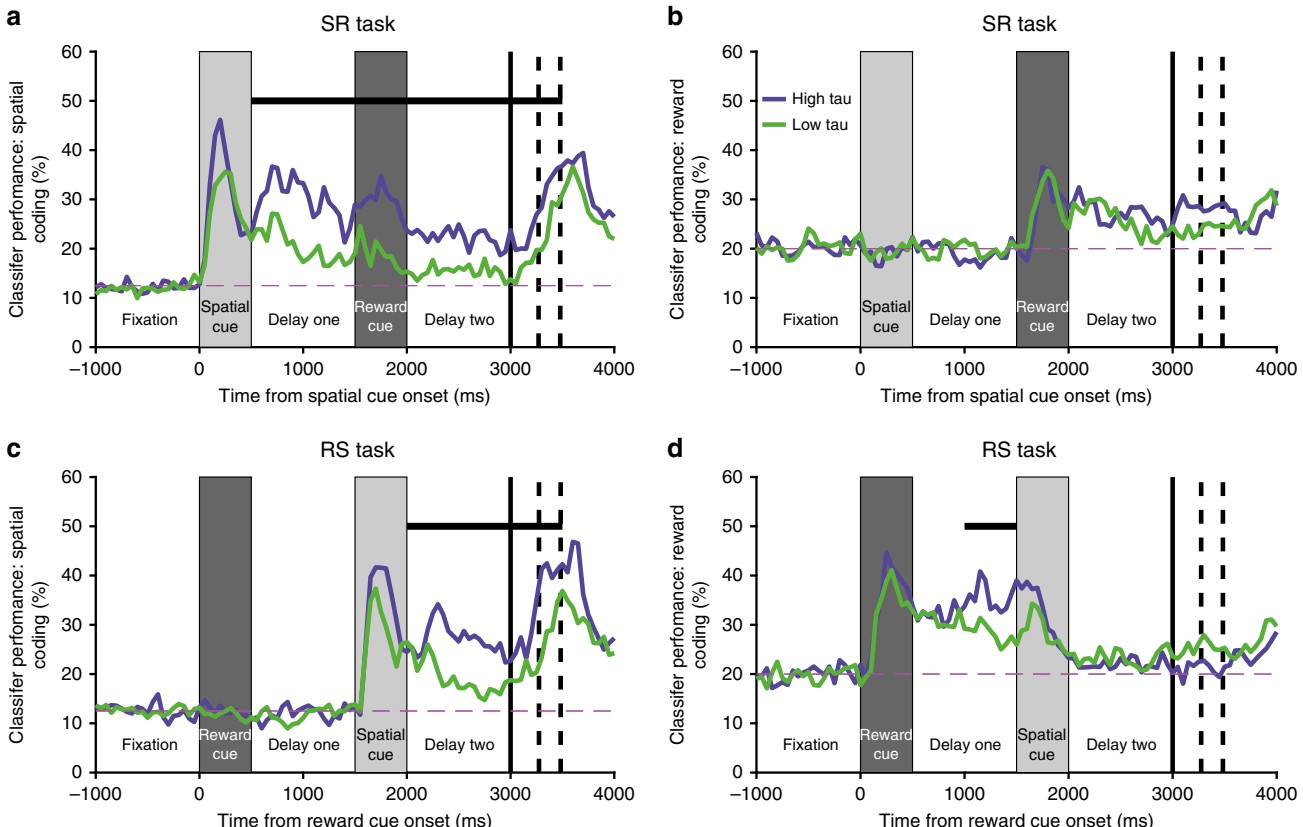

**Fig. 3** Ventrolateral prefrontal neurons with higher resting time-constants maintain reward and spatial information across delays. The mean performance of classifiers (1000 permutations, Methods) trained to decode space (**a**, **c**) and reward size (**b**, **d**) is calculated for two subpopulations of ventrolateral prefrontal cortex (VLPFC) neurons subdivided by resting time-constant. For spatial coding on the SR-task (**a**), the subpopulation with higher time-constants has a stronger representation during the first delay (first 500 ms $p = 0.00313$, second 500 ms $p = 0.00041$, Bonferroni-corrected bootstrap tests; Methods), whilst the reward cue is on screen ($p = 7 \times 10^{-5}$), during delay-two (first 500 ms $p = 0.00824$, second 500 ms $p = 0.01736$), and during the early response period ($p = 0.0355$). The high time-constant population also has stronger spatial coding in delay-two of the RS-task (first 500 ms $p = 0.03237$, second 500 ms $p = 0.0003$, **c**) and in the early response period ($p = 0.00461$). Ventrolateral PFC high time-constant neurons also code reward more strongly during delay-one of the RS-task (Second 500 ms $p = 0.0330$, **d**). Horizontal black bars represent a significant difference between the high and low time-constant subpopulations (Bonferroni-corrected bootstrap test for 10 non-overlapping 500 ms epochs; Methods, $p < 0.05$). The dashed magenta line represents chance level classifier performance. The first solid vertical line signifies when subjects were cued to respond. The first and second dashed vertical lines represent the average timing of the subjects' saccade and the onset of reward, respectively

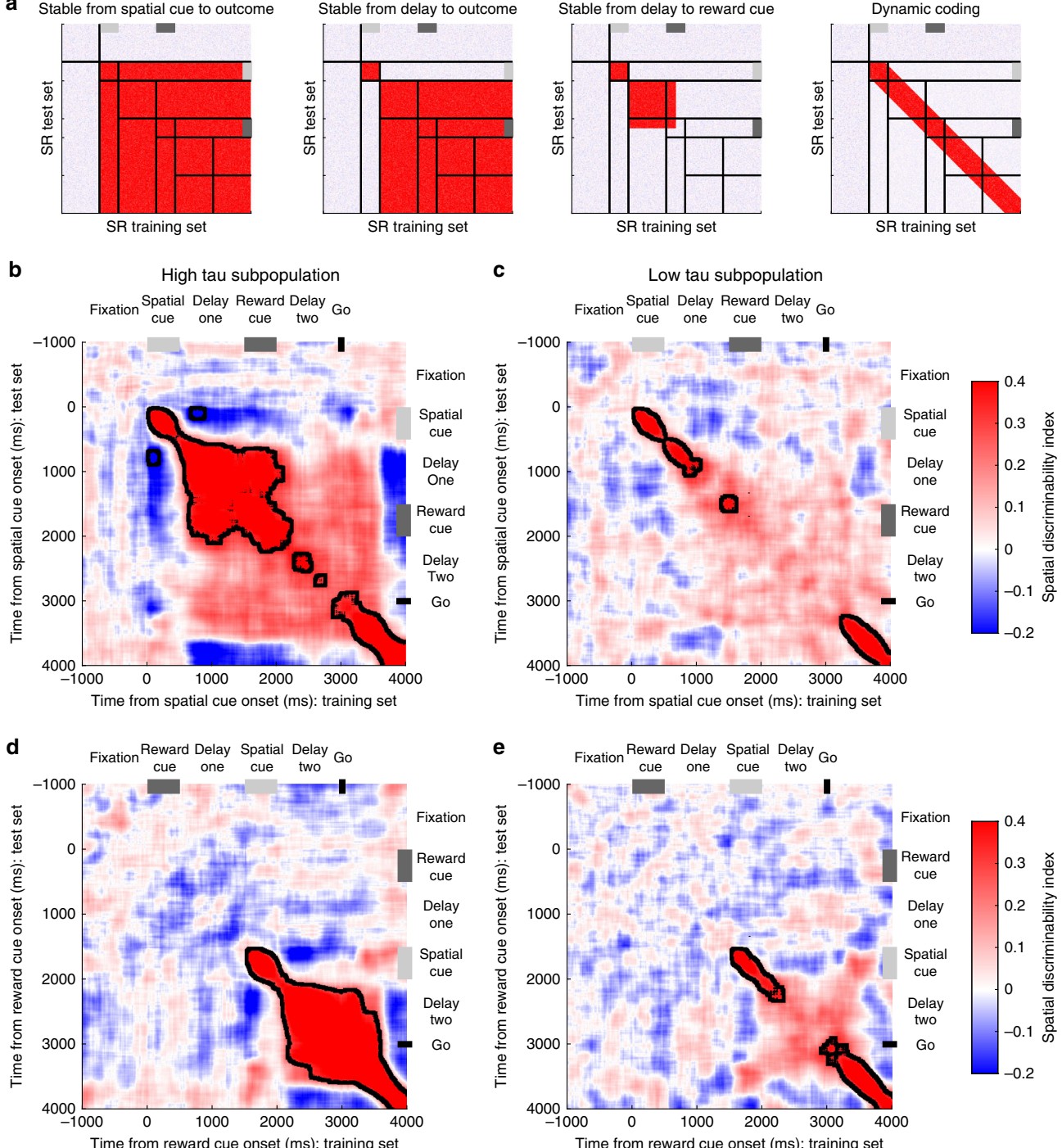

**Fig. 4** Cross-temporal dynamics of spatial selectivity by high and low time-constant populations. **a** Schematic representing cross-temporal dynamics of different working-memory codes on SR-trials. Each pixel represents how well spatial location can be discriminated when using half of the trials at one timepoint as a training set (X-axis), and the other half of trials at a separate timepoint as a test-set (Y-axis). Off diagonal, the plot indicates the stability of any spatial coding across time. In the first exemplar, stable spatial coding is evident across the trial, as data from any timepoint after cue presentation can be used to decode the remembered spatial location at any other timepoint. The second exemplar is similar, but this stable state is only established following a transient dynamic phase where the cue is initially encoded. The third exemplar shows that this stable state is established during the initial delay—but collapses after the reward cue is presented. The final exemplar shows that spatial location is coded throughout the trial (heat on the diagonal), but that this code is not stable across time. **b–e** Cross-temporal decodability of spatial location is plotted for high (**b**, **d**) and low (**c**, **e**) time-constant VLPFC populations on SR (**b**, **c**) and RS (**d**, **e**) trials. The high time-constant subpopulation has greater stability of spatial coding: the off-diagonal elements are warm, meaning that the same population code persists throughout the delay epoch following the spatial cue. Despite this stability, there is a negative correlation between the cue period and the delay indicating a reversal of spatial tuning between these epochs. In SR-trials, a stable state is reached during delay-one, but this is disrupted by the presentation of the reward cue, and there is only a weak non-significant cross-temporal generalisation between delay-one and delay-two. A dynamic, rather than stable, representation of space returns around the time of the go cue. In the low time-constant population, coding is always dynamic, so no stable state is established. Black lines encircling areas of strong coding indicate significant cross-temporal stability ($p < 0.05$, cluster-based permutation test, Methods)

encoding state and the relationship between neural codes across the trial (Methods)[30,35]. To study cross-temporal generalisation of task selectivity, a classifier is trained at one timepoint ($t$) and tested at a different timepoint ($t + \delta$). If there remains a strong correlation between the test and training set at two distinct timepoints, selectivity generalises across the period between the two timepoints. By using all $n$ timepoints as training and test sets, an $n \times n$ correlation matrix can be constructed.

The resulting pattern of generalisation can distinguish between different WM models, as indicated by the exemplars in Fig. 4a. The first example shows a stable attractor model on SR-trials[7]. Soon after spatial cue onset, a stable state of network activity forms specific to each spatial location. This pattern of activity generalises (i.e., the off-diagonal regions) throughout the time the stimulus remains in WM (illustrated by red colour from stimulus presentation onwards). A revised stable subspace version of this model incorporates a dynamic component during the cue period, with stability only established in the delay period (Exemplar 2)[12]. In this version, spatial coding during cue presentation does not generalise to later periods in the trial, but a stable attractor is formed around the time of stimulus offset. A third exemplar shows what may happen if this stable subspace were to be disrupted by the presentation of the reward cue. A final example shows a purely dynamic coding model[14,35], whereby dynamic on-diagonal selectivity maintains an active representation of spatial information across time, but this never reaches a fixed point of stable network activity (i.e., lack of off-diagonal shading).

The pattern produced by the VLPFC high-tau subpopulation exhibited elements consistent with both stable and dynamic coding (Fig. 4b, d)[12,40]. Coding from the spatial cue period was not positively correlated with the subsequent delay, consistent with dynamic activity during the initial encoding phase. Surprisingly, neural activity was anti-correlated between the cue and delay-one ($p < 0.0001$; cluster-based permutation test, Methods), suggesting the way the network encodes spatial information reverses between cue presentation and delay. This selectivity pattern reversal was also evident in VLPFC reward coding, but not present in any other PFC area (Supplementary Figure 4).

Despite this dramatic reversal of selectivity from the cue to delay periods, a stable state of cross-temporal generalisation was established in the high-tau subpopulation during delay-one, which was sustained through the reward cue epoch (Fig. 4b; $p < 0.0001$; cluster-based permutation test). This finding is consistent with the VLPFC high-tau subpopulation demonstrating attractor-like WM activity in classical tasks without intervening stimuli[1,7,12]. However, the cross-generalisation of maintained spatial information was disrupted during delay-two on SR-trials, and there was no prolonged significant generalisation between the activity in delay-one and delay-two (Fig. 4b). The fact that network activity in the VLPFC high-tau subpopulation is dynamic at cue presentation, then exhibits a reversed stable state of generalisation which is disrupted following the reward cue, suggests VLPFC network activity is not performing the function of a conventional attractor for spatial WM[7].

Compared with high-tau cells, the VLPFC low-tau subpopulation had more transient dynamics (Fig. 4c, e). Although there is weak on-diagonal selectivity, this does not extend off-diagonal, consistent with dynamic coding. The spatial selectivity in the high-tau subpopulation was significantly more stable over time during the post-stimulus delay and shortly after (SR-trials $p < 0.0001$, RS-trials $p = 0.0388$; cluster-based permutation test; Supplementary Figure 5). In summary, of all the PFC subpopulations examined, only the high-tau VLPFC subpopulation formed a stable spatial mnemonic representation, but the additional task

element of a salient reward cue showed this state to be inconsistent with current attractor models.

**Anti-correlation between Cue and delay period activity**. Recent work suggests that stable population activity can co-exist alongside strong temporal dynamics during the initial encoding phase[12]. This can occur if the mnemonic representation is established at the time of the cue but is accompanied by a transient, orthogonal pattern of activity. These results would appear inconsistent with the reversal of spatial coding we observed in the VLPFC high-tau population between cue presentation and delay. To further examine this issue, we correlated activity within the VLPFC high-tau subpopulation across time within each condition (Fig. 5a, Methods)[12]. This revealed a strong positive correlation across the whole trial, including between cue and delay periods (Fig. 5a). This suggests that within a given spatial location, VLPFC high-tau firing rates were stable and correlated across the trial (as opposed to the instability and reversal of mnemonic coding evident in Fig. 4). Whilst this may be taken as evidence against a reversal of selectivity patterns, we reasoned this positive correlation may be largely driven by the intrinsic firing rates of the neurons (e.g., a neuron which is high firing during the cue may also be higher firing during the delay even if it is modulated across the trial). By demeaning activity (Methods) across conditions for each neuron and repeating the analysis, we revealed an anti-correlation in the activity of high-tau VLPFC neurons between the spatial cue and delay periods (Fig. 5b). The high cross-trial correlations observed in Fig. 5a are therefore likely driven by neurons possessing relatively consistent firing across the trial.

To further examine the stability and pattern of spatial selectivity, we employed principal component analysis (PCA). Previously, PCA revealed a mnemonic subspace that was stable from cue onset through the delay[12]. The mnemonic subspace was defined by time-averaging delay activity for each stimulus condition for each neuron and running PCA across conditions (conditions × neurons matrix). Projecting data from the cue period into this subspace still enabled decoding of spatial position, supporting the proposal that the stable delay activity is already established during cue presentation[12].

We tried to replicate this approach in the high-tau VLPFC subpopulation (Fig. 5c–e, Methods), by defining the subspace based upon time-averaged delay-one activity in SR-trials. We then projected neural firing from across the trial onto the first two principal axes (Fig. 5c, d). If the mnemonic representation is stable, all traces should be fairly fixed and separable across time (as in ref. 12 Fig. S3). During delay-one, there was a stable representation of mnemonic information, as all conditions are separable within this subspace. The representation of space is also shown to be geometrically consistent with the spatial environment, with the activity for nearby spatial locations clustered in the subspace. However, supporting our previous analyses, projecting neural activity from the cue period into the subspace did not lead to a reliable spatial code. Remarkably, the spatial conditions were separable in the cue period, but in the opposite direction to that observed during delay-one. This pattern was also replicated for reward coding on RS-trials, suggesting this reversal is a general pattern of VLPFC coding between cue and delay periods, and not limited to spatial selectivity (Supplementary Figure 6). To quantify the reliability of the SR delay-one subspace, we calculated the variance explained by projecting data at each timepoint (Fig. 5e). Unlike previous findings[12], the mnemonic subspace in the delay explains only a small proportion of variance during the cue period.

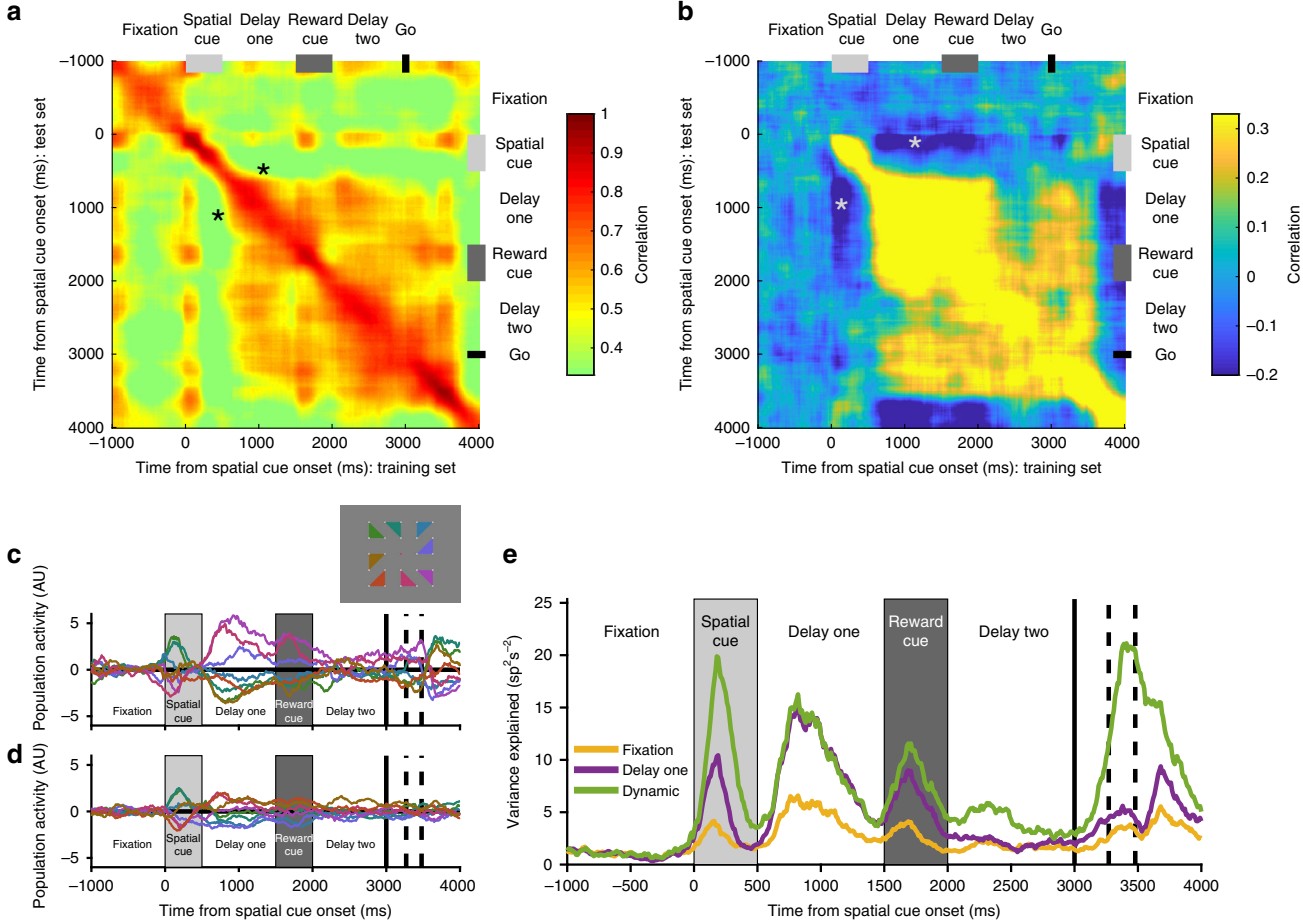

**Fig. 5** VLPFC high time-constant population reverses its spatial coding between cue presentation and the subsequent delay. **a** Within-condition correlation of neural firing across time for SR-trials (Methods). All bins are positively correlated with each other, suggesting neural firing is stable across time. Note positive correlation between cue period and delay (asterisk). **b** Within-condition correlation analysis where activity for each neuron was demeaned across each of the spatial locations (Methods). There now exists a negative correlation between the time of the spatial cue presentation and delay-one (asterisk). **c**–**d** Reversal of VLPFC high time-constant spatial tuning between cue and delay. A mnemonic subspace was defined with time-averaged delay-one activity. The across-trial firing for each condition was projected back onto the first (**c**) and second (**d**) principal axes of this subspace. While the conditions remain well-separated on both principal axes during delay-one, the subspace does not generalise well into delay-two as activity from the different conditions converges. At the time of the spatial cue, the conditions appear separable, but in the reverse configuration from that during the delay. The inset shows the geometric location of each spatial location that appeared on the screen. **e** The stimulus variance captured by three different subspaces is displayed. The fixation subspace is defined by time-averaged activity in the 1000 ms before cue presentation. This should represent a chance level amount of variance explained. The delay-one subspace is defined by time-averaged activity from 500 to 1500 ms after cue presentation. The dynamic subspace is defined separately at each individual time point. The dynamic subspace explains a much greater amount of variance during the cue period, illustrating that there is little consistency in the activity patterns between spatial cue and the delay epochs. However, the delay-one subspace captures as much variance as the dynamic subspace during delay-one, suggesting the VLPFC high-tau population activity has settled to a stable state by this point

In short, we find little evidence that the VLPFC high-tau subpopulation forms a stable subspace maintaining information from cue onset through the delay. Rather, the population geometry reverses its selectivity pattern for both reward and spatial information between the cue and delay periods, before forming a stable subspace that maintains WM-related information across the initial delay until the subsequent interfering cue.

**Cross-task generalisation**. Thus far we have demonstrated that only high-tau VLPFC neurons exhibit stable cross-temporal generalisation of mnemonic information. We next explored whether there was cross-task generalisation between SR- and RS-trials. Previous studies have demonstrated task-specific PFC activity to identical stimuli when they cue a different response[41,42]. However, whether the pattern and stability of population activity depends on the order in which identical information (cueing the same response) is received remains unknown. To explore this, we used data from SR-trials as a training set, and data from RS-trials as a test set. This analysis allowed us to examine, for example, whether the population pattern for spatial selectivity that emerges in delay-one of SR-trials (Fig. 4b) is similar to the population pattern for spatial selectivity in delay-two of RS-trials. This analysis also allowed us to test whether the population pattern in delay-two was similar across both trial types; at this point in the trial, the subjects have processed the same information and are required to prepare the same response.

Figure 6a depicts three possible exemplars to interpret the cross-task generalisation analysis. We performed the analysis on all recorded VLPFC neurons. The activity pattern of this population was primarily consistent with stimulus locked generalisation (Fig. 6b); there is strong cross-task generalisation between spatial cue presentation and the subsequent 1-s delay

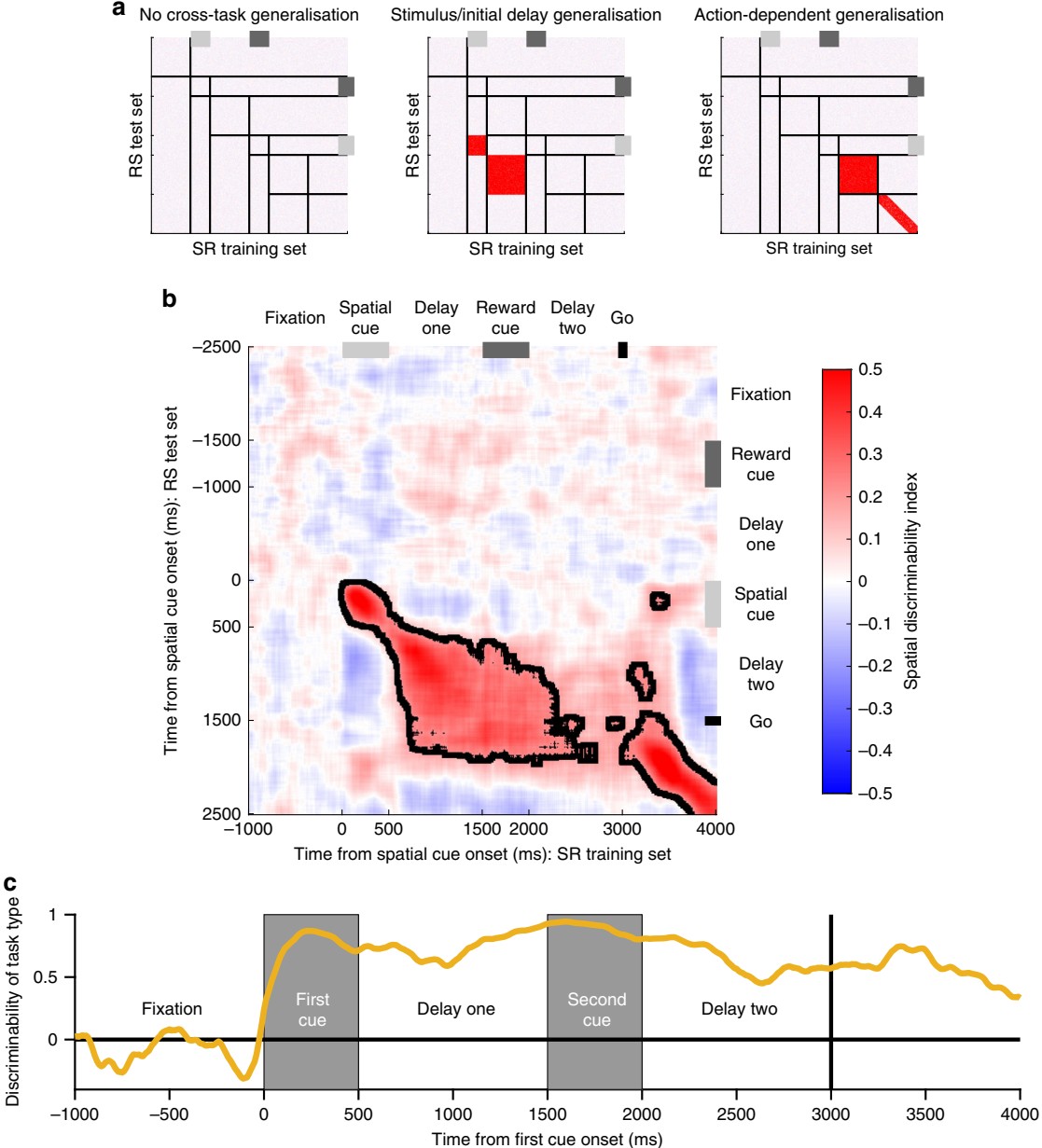

**Fig. 6** Cross-generalisation of working memory activity across-trial types. By using data from SR-trials as a training set for a classifier, and data from RS-trials as a test set, the generalisability of spatial coding across task types can be studied. **a** Exemplars of how population activity may generalise across-trial types. If there is no cross-task generalisation, spatial position cannot be decoded from neural activity on the other trial type. As VLPFC has spatial coding on both trial types (Fig. 2), if there is no cross-task generalisation this would mean there are multiple network patterns of spatial selectivity capable of supporting correct performance. If there is stimulus-locked generalisation, spatial position can be decoded by activity from the other trial type; however, it is relative to cue presentation so the decoding is displaced off the diagonal. In this scenario, spatial location could be readout identically across-trial types using activity post-stimulus presentation (red colour on heatmap), but because spatial selectivity on SR-trials is disrupted by the reward cue (Fig. 4), distinct readout weights would be required at the time of response. If there is action-dependent generalisation, neural activity generalises in delay-two and response epochs as subjects prepare and execute their saccade. This may occur if a different route through neural state space is taken on the two trial types, but the routes converge and the same common trajectory is reached by delay-two. **b** Cross-generalisability in VLPFC is primarily locked to the presentation of the stimulus. Spatial position cannot be decoded from activity during delay-two, implying distinct population codes on the two trial types in the delay immediately prior to response. Only once the action is initiated (at the go cue), does a cross-trial generalisation appear on the diagonal. Dashed lines encircling areas of strong coding indicate a significant cross-generalisable stability (p < 0.05, cluster-based permutation test, Methods). **c** Decoding task type. The task the subjects are performing can be accurately decoded from VLPFC neural activity, throughout the trial. This is particularly important during delay-two, as at this point the subject has been exposed to the same visual stimuli, just in reverse order

(cue-period p < 0.0001; delay period p < 0.0001; cluster-based permutation tests). There is then little cross-task generalisation in delay-two, indicating distinct activity patterns in this epoch between the two tasks. We confirmed a strong representation of trial type during delay-two using a separate decoding algorithm,

which discriminated activity between trial type (Fig. 6c, Methods). These results indicate that a different set of read-out weights for WM of spatial location would be required from VLPFC activity for correct performance on the two trial types, implying independent task-specific neural states can support WM.

**Single neurons switch between reward and spatial coding**. Thus far, the results suggest a heterogeneous and primarily dynamic account of WM activity within the PFC population. To examine the underlying pattern of this population heterogeneity, we analysed single-neuron selectivity for different task features. This analysis explored how strong and sustained WM selectivity patterns were in individual neurons[8,40], how these WM representations were affected by the presentation of a second salient

cue, and whether neural activity in delay-two encoded a combination of task variables[23,24].

To quantify single-neuron encoding of both reward and spatial information, we ran a separate Kruskal–Wallis test for space and reward at each timepoint (Fig. 7a–h, Methods). On SR-trials, a large proportion of VLPFC neurons were selective for spatial location during cue presentation or delay-one (Fig. 7a). These neurons had heterogeneous onset latencies and most were transiently selective, as

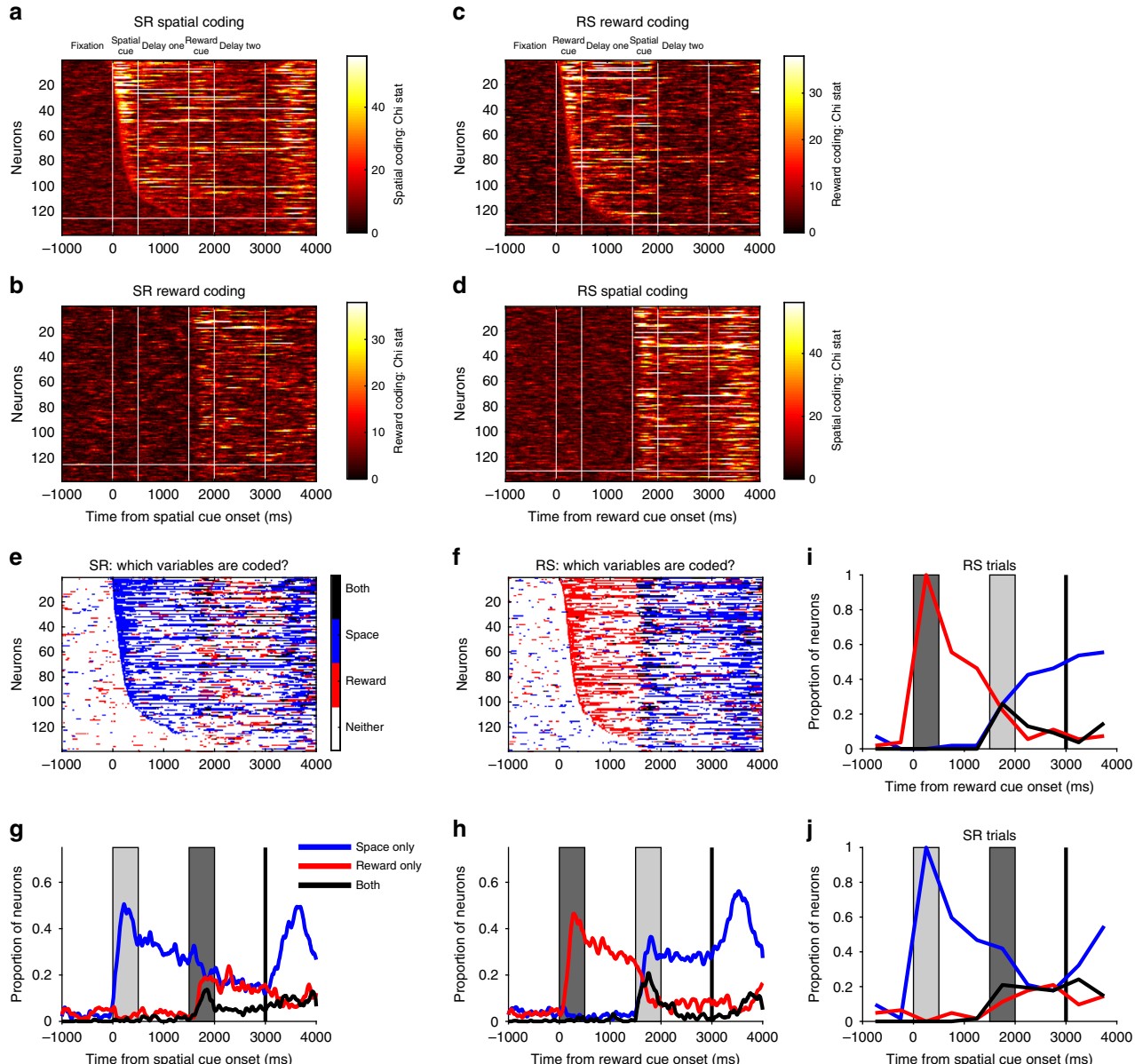

**Fig. 7** Flexibility of single-neuron selectivity. **a–b** SR-trials: Single neuron coding. **a** The heatmap shows the spatial coding of individual ventrolateral PFC neurons; each row of the matrix represents single-neuron selectivity. Neurons are sorted by their latency for spatial encoding; all neurons above the horizontal white line were selective for space either during cue presentation or delay-one. For many of these cells, selectivity is transient; few code space across extended periods of the trial. **b** Reward coding of individual ventrolateral PFC neurons. A large proportion of the neurons which were selective for space subsequently become selective for reward at cue two/delay-two (neurons are sorted in the same order as in **a**). **c–d** RS-trials: Single neuron coding. Neurons are now sorted by their latency for reward encoding (**c**), with all neurons above the white line selective during cue presentation or delay-one. Reward encoding is primarily transitory in nature. **d** This heatmap shows that many of the neurons initially coding reward go on to code the spatial location when this cue is presented. **e–f** The task variables significantly (Methods) encoded by each neuron at each timepoint are plotted for SR (**e**) and RS-trials (**f**). **g–h** Fraction of neurons selective for either or both task factors across SR (**g**) or RS-trials (**h**). Presentation of the second stimulus reduces the number of neurons selective for the initially presented cue. **i–j** Switching of selectivity across a trial. Neurons are included in this analysis if they were selective during the presentation of the first cue. The selectivity pattern of these neurons is profiled across time. On SR-trials (**j**), only a minority of cue selective neurons retain an exclusive representation of space across the entire trial; many neurons gain reward coding, some at the expense of spatial selectivity, and others in addition to this. On RS-trials (**i**), a similar dynamics are observed

opposed to sustaining a spatial representation across time. Strikingly, many of these spatially selective neurons subsequently coded reward size later in the trial (Fig. 7b). A similar result was also observed on RS-trials, where many reward-selective neurons (Fig. 7c) subsequently encoded spatial location (Fig. 7d). This is consistent with the VLPFC population analysis (Fig. 2) showing that the most recently presented stimulus is primarily encoded, as opposed to the task-relevant spatial information necessary for correct performance. This result can be explained by single PFC neurons being involved in multiple distinct cognitive functions[23], as

opposed to different subpopulations representing different task-related factors becoming active at different stages of the trial.

The ability of PFC neurons to encode both reward and spatial information may highlight neuronal flexibility, or the capability to code multiple factors concurrently. Fig. 7e–h characterise the proportion of neurons simultaneously coding spatial and reward information. During the presentation of the second cue, some neurons appeared to multiplex reward and spatial information. To establish the nature of this mixed selectivity, we ran a two-way ANOVA to explore any interaction effects (Fig. 8, Methods). It

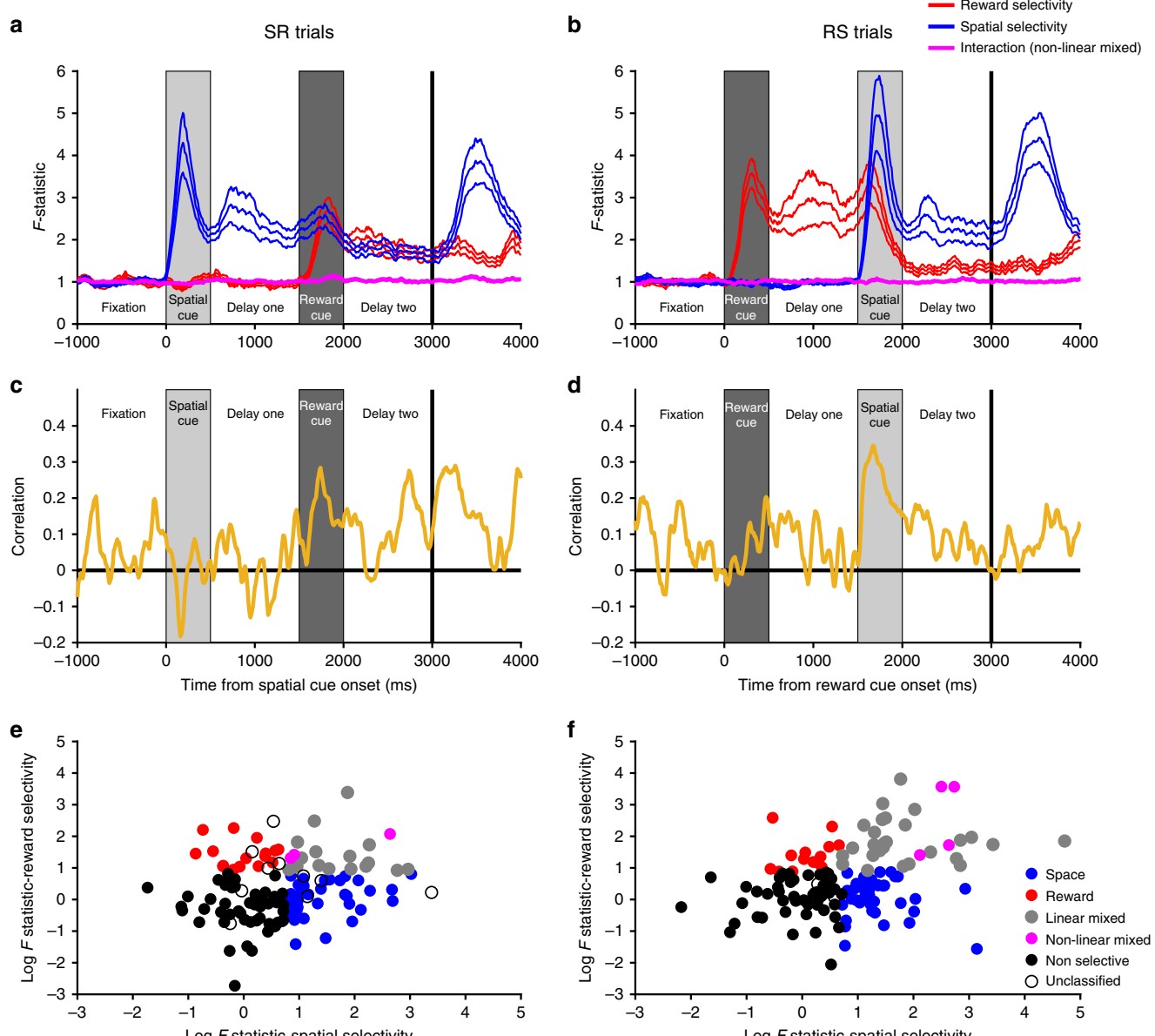

**Fig. 8** Absence of non-linear interactions between reward and spatial selectivity. **a–b** The mean population F-statistics from a sliding two-way ANOVA with an interaction term are plotted (±standard error of the mean) for SR (**a**) and RS (**b**) trials. The interaction term between both factors does not change from that during pre-trial fixation. **c–d** Sliding spearman correlation between reward and spatial selectivity F-statistics for SR (**c**) and RS (**d**) trials. On both trial types, there is a positive correlation at the time of cue two, indicating linear mixed selectivity. **e–f** Spearman correlation of spatial and reward coding F-statistics during cue two for (**e**) SR-trials; (**f**) RS-trials. To complement the above analysis, we performed a single spearman correlation between the raw F-statistics from a 2-way ANOVA of spike-counts during the entire cue two period. There was a significant positive correlation on SR-trials ($r = 0.175$, $p = 0.0394$) and RS-trials ($r = 0.311$, $p = 2.08 \times 10^{-4}$). Each dot represents a neuron appropriately coloured. Space and reward neurons were required to have only one significant main effect ($p < 0.05$) and a non-significant interaction ($p > 0.05$). Linear mixed neurons were required to have two significant main effects ($p < 0.05$) and a non-significant interaction ($p > 0.05$). Non-linear mixed neurons were required to have two significant main effects ($p < 0.05$) and a significant interaction ($p < 0.05$). Non-selective neurons had no significant effects ($p > 0.05$). A very small number of neurons did not meet these criteria so were unclassified. F-statistics have been log-transformed for illustrative purposes

could be that neurons code both factors with a non-linear interaction[20,23], exhibiting a different pattern or degree of spatial coding at each reward level. Alternatively, both factors could be coded simultaneously without an interaction[43] (e.g., similar pattern of spatial selectivity for each reward level resting upon a different baseline firing rate). We found little evidence for non-linear mixed selectivity (Fig. 8a, b). Instead, there was a positive correlation between selectivity for space and reward at the time of the second cue (Fig. 8c–f), implying most neurons that exhibit mixed selectivity for multiple factors do so as a linear combination.

This flexibility of single-neuron selectivity patterns appears inconsistent with more traditional accounts of PFC function during WM. We next quantified the proportion of neurons exhibiting stimulus-specific selectivity throughout the trial (Fig. 7i, j, Methods). On SR-trials, this showed that only a minority (17.74%) of the spatially selective neurons at cue presentation are selective for spatial location alone by the end of delay-two. Thus unlike classical notions of WM being supported by sustained selectivity[1,2], our results suggest single neurons do not maintain sustained WM correlates[9], at least not in cases where other task-relevant or salient information compete for neuronal representation.

## Discussion

Here, we used an oculomotor-delayed-response task with an intervening reward cue to test whether WM is subserved by persistent activity in single neurons or by dynamic activity across a neural population. This task design allowed us to specifically contrast these different hypotheses of WM coding. A recent cortical attractor model of WM would suggest a dynamic cue-related response followed by a stable state of fixed network activity specific to the mnemonic stimulus[12]. This model would predict that if changes in this stable state were induced by a subsequent salient stimulus, this would compromise the WM representation. This constraint does not apply to WM models that do not rely on stable network states. Of the four PFC subregions examined, mnemonic selectivity spanning the entire delay was evident only in VLPFC neurons, and this was present only in the subpopulation of neurons with high time-constants. Within these VLPFC neurons, the pattern of both reward and spatial selectivity reversed from the cue to delay, where it then became stable and generalised across time. However, once the reward cue was presented, spatial selectivity was largely quenched and instead the VLPFC population switched to coding the salient reward information. These results demonstrate that high-tau VLPFC neurons are capable of stable selectivity that could serve WM functions, but that in contexts where multiple behaviourally relevant stimuli are available, VLPFC neurons flexibly code the focus of current attention[24,34,36].

Both attractor[7,12,18] and synaptic models[13] of WM stress the importance of a recurrent network architecture. By using the decay of autocorrelation of spiking activity during a fixation period as an unbiased metric of intrinsic persistent activity, we demonstrate that neurons with higher time-constants (taus) are more likely to exhibit WM-related selectivity, but only in the VLPFC population. The VLPFC high-tau subpopulation had stable selectivity during the initial delay period following stimulus offset, whereas the low-tau subpopulation exhibited dynamic coding. Importantly, any distinction between the high and low-tau VLPFC subpopulations was only evident during this mnemonic phase, ruling out the possibility that high-tau cells are simply more task-selective. However, ACC, the PFC region where the longest taus were observed, did not display prolonged selectivity for either reward or spatial information. This suggests that a

neuron's selectivity pattern is constrained by both the functional anatomy of its brain region and its tau. High-tau neurons in ACC may perform complex functions across longer timescales (e.g., integration of information across trials[44]) than our task was designed to investigate.

Our finding that high-tau VLPFC neurons exhibit stable WM correlates builds upon recent work showing PFC neurons with higher taus have a greater role in decision-making and the maintenance of relevant information over time[30], highlighting a broader role for high time-constant neurons subserving extended cognitive processes. These findings appear supportive of theories proposing that cortical attractor networks fulfil WM functions[7,12]. However, we also observed several features of the data which suggest VLPFC activity is incompatible with current attractor models. Firstly, we showed that VLPFC reverses both its spatial and reward tuning between cue presentation and the subsequent delay. Previous studies have shown that cue and delay dynamics are distinct[12,35,40], but our discovery that the tuning geometry reverses between cue and delay is novel. This finding is also inconsistent with a stable subspace spanning both cue presentation and memory[12]. The inverted tuning geometries may reflect a mechanism to dissociate stimuli currently in the environment and those held within memory[45], or alternatively, a mechanism to load information into WM from an initial state of dynamic sensory encoding.

By probing the effect of a salient reward cue on the stability of mnemonic representations, we were able to further test whether cortical attractors in PFC provide a mechanism for distractor-resistant WM. It was shown that the intervening reward cue quenched the WM selectivity pattern in the VLPFC population. A recent report similarly showed that a behaviourally irrelevant distractor morphed spatial selectivity of PFC neurons[20]; but the distractor was not encoded, presumably because the subjects dismissed it as inconsequential information. The PFC population activity, although morphed with respect to activity pre-distraction, could therefore continue to maintain a strong mnemonic representation. In contrast, the reward cue in our paradigm had behavioural relevance, and reward anticipation commonly modulates PFC neurons[21,46–51]. We found that many neurons holding the spatial representation flexibly switched to code the reward. This suggests that different neural mechanisms may be required to maintain WM when a salient stimulus also carries behavioural relevance and activates neurons across PFC. This WM mechanism seemingly eludes current attractor models, which predict distractor-resistant spatial selectivity within PFC[12,52].

The dynamic switch of VLPFC activity to coding the behaviourally relevant reward cue provides further evidence that PFC neurons can be tuned to multiple diverse cognitive factors, and that they can flip between them within the course of a trial[23,40,53]. It also suggests previous studies concluding PFC neurons are resistant to distraction by intervening cues during a WM delay do not generalise to more behaviourally salient stimuli[25–27]. Here, we used a reward-predictive cue presented at the fixation spot, as opposed to a peripherally flashed target[26] or stimulus[20,25], which is irrelevant to the task. Our results concur with a recent report of distractor encoding within PFC when a centrally presented stimulus interrupted a WM delay[28].

Inverted tuning between cue and delay, a weakening of a stable mnemonic representation by a salient reward cue, and neurons flexibly encoding both factors, all suggest VLPFC activity is incompatible with existing cortical attractor models[12]. There are several possible interpretations of the WM activity we observed across PFC. Although WM-related activity and WM deficits following brain damage are both most commonly associated with LPFC[4,5,54], it is conceivable that classical distractor-resistant stable activity was present in a PFC region we did not sample.

However, we sampled a large expanse of LPFC including both banks of the PS (areas 9/46d, 9/46v), and several millimetres of cortex both dorsal (area 9) and ventral (areas 45A, 45B, and 47/12) to PS, as well as parts of the medial (ACC) and ventral (OFC) PFC. Mnemonic activity has been observed in other brain regions, such as the parietal cortex[55,56]. However, this activity is more sensitive to distraction[26,57], and parietal inactivation produces comparatively modest WM deficits relative to PFC, suggesting a less prominent role in WM processes[6,58,59]. A further possibility is that we may have missed stable, persistent activity in PFC because of a more anterior recording location than previous studies[60]. We also consider this interpretation unlikely. Recent studies recording more posteriorly in LPFC including the frontal eye field have shown that mnemonic spatial selectivity is not stable when either multiple mnemonic stimuli are subsequently presented or a distractor appears[20,61]. Instead, we note that the vast majority of tasks reporting stable coding do so during a delay period where there is only one mnemonic representation to be maintained and no intervening stimuli[2,3,12,40]. Had our study similarly terminated at the end of delay-one on SR-trials (Fig. 4b), our results would be highly consistent with findings of these tasks[12]. Crucially, without the presentation of an additional salient cue, both the most recently presented stimulus and the posited locus of the subject's attention are confounded with WM[36]. Our findings suggest stable mnemonic representations are present in PFC, specifically in high-tau VLPFC neurons, but that these neurons can also flexibly switch which information they encode as other behaviourally relevant variables compete for the subject's attention.

Relative to VLPFC, DLPFC had weak and short-lived coding of spatial location (although more prolonged than ACC or OFC). This is perhaps surprising, as some hypotheses propose DLPFC primarily maintains WM for spatial locations, whereas VLPFC maintains WM for object identities[60,62]. However, strong connections exist between parietal cortex and VLPFC[63,64], which could provide spatial information, and VLPFC receives input from high-level visual areas in the temporal lobe[65], which could provide object-related information. Perhaps consequently, other reports observe equivalent or stronger spatial selectivity ventral to the PS[24,34,36,66]. These studies all required subjects to attend to spatial information in addition to other task features (e.g., reward information, object identity, and task rule). Thus, it is possible that spatial selectivity may shift towards VLPFC in contexts where flexible allocation of attention to multiple task features is required.

An important question therefore remains how WM is achieved on this task. The residual level of dynamic VLPFC spatial coding may be sufficient for WM performance. Another possibility, although not directly verifiable with our data, is that a PFC region maintains a representation of the mnemonic stimulus in an activity silent state[13,14]. Alternatively, PFC may be essential for setting up a stable mnemonic representation during the initial delay, but if this mnemonic information specifies a response that could be prepared, this information could then be transmitted to oculomotor regions to prepare a saccade. This would be akin to activity in premotor regions for reaching movements[67]. It would be worthwhile to explore these ideas in a task where the contents of WM were independent of the response[8,10,41], and investigate the impact of a salient intervening stimulus on the WM representation. Regardless, our data are incompatible with PFC maintaining WM in cortical attractor networks throughout an oculomotor-delayed-response task, when the delay is interrupted with behaviourally relevant information. This provides novel neurophysiological evidence that stable activity states within PFC may be more tightly associated with the most recently presented behaviourally relevant stimulus, rather than the contents of WM.

## Methods

**Subjects and neurophysiological procedures**. Neurophysiological procedures and task design have been reported, as this is a re-analysis of previously published data[33,34]. In brief, two male rhesus macaques (Macaca mulatta) aged 5–6 years served as subjects. All procedures were in accord with the National Institute of Health guidelines and the recommendations of the University of California at Berkeley Animal Care and Use Committee. Single neurons were recorded from four brain regions across PFC (Fig. 1b): ACC (areas 9 m, 24c, $n = 198$), DLPFC (areas 9/46d, 46d, 46v, 9/46v $n = 205$), VLPFC (areas 8AV, 45A and 45B, 47/12, $n = 139$) and OFC (areas 11, 13, $n = 152$). The brain areas associated with each recorded neuron were estimated using a macaque monkey brain atlas[68], following physiological validation of anatomical landmarks (i.e., sulci) to reconstruct the precise locations of all recorded neurons[33,34]. For this report, we classified all neurons recorded inside the PS, and within ~2 mm of lateral surface both dorsal and ventral to the PS (in areas 9/46d, 9/46v), as DLPFC. All neurons in brain areas classified as VLPFC were thus located ~>2 mm ventral to the PS. No statistical methods were used to predetermine sample sizes, but our sample sizes are similar to those reported in previous publications. We randomly sampled neurons and did not attempt to pre-select neurons based on responsiveness to enable a fair comparison of neuronal properties between different brain regions.

**Task**. A detailed overview of the task structure has been described elsewhere[33,34]. We monitored eye position and pupil dilation during the task using an infra-red system (ISCAN). Subjects first fixated a central cue for 1000 ms before two cues were presented sequentially, each for 500 ms, each followed by a 1000 ms delay. One of the cues was a spatial location that the subject had to hold in WM, and the other indicated how much reward the subject would receive for correct performance of the trial. We used 24 different spatial positions and two different reward-predictive picture sets, each cue indicating one of five reward levels (Fig. 1a). The reward cues displayed in Fig. 1a are similar, but not identical to those used in the study. The original images are available elsewhere[33,34]. The 24 spatial targets were regularly distributed in a 5 × 5 matrix centred at the fixation spot, with each position separated by 4.5° of visual angle. The 24 spatial targets were collapsed into eight locations forming triangles (Fig. 5C, **inset**) to allow for sufficient trials for the decoding analyses. For the error-trial analysis (Supplementary Figure 1), the 24 spatial targets were collapsed into four locations by combining tessellating pairs of triangles into rectangles. On SR-trials, the spatial position was shown first followed by the reward cue, whereas on RS-trials the cues were presented in the reverse order. If subjects maintained fixation through both the cue and delay periods, the fixation cue changed colour and the subject could initiate a saccade to the remembered spatial location (Fig. 1a). If the saccade terminated within 3° of the remembered target and was maintained in this location for 150 ms, a reward was delivered and the trial was recorded correct. Trials where fixation was maintained but the saccade failed to terminate in the remembered location were recorded as errors. We counterbalanced all spatial positions and reward levels, and the two trial types were randomly intermingled.

**Time-constant analysis**. Single-neuron activity during a 1000 ms fixation period was used to assign time-constants (Fig. 1c, d)[30]. Single unit responses were time-locked to the onset of the fixation period of successfully completed trials. Fixation-period rasters were divided into 20 discrete, successive 50 ms bins. The spike count for each neuron within each bin was calculated for each trial. Pearson's correlation-coefficient was used to compute the across-trial correlation of spike-counts between all of the bins. For each single-neuron, this produced an exponential decay when autocorrelation was plotted as a function of time lag between bins (as in Fig. 1d). The decay of the autocorrelation was fitted to the data using the following equation:[31]

$$R(k\Delta) = A\left[\exp\left(-\frac{k\Delta}{\tau}\right) + B\right] \qquad (1)$$

In which $k\Delta$ refers to the time lag between time bins (50–950 ms) and $\tau$ is the decay time-constant (in ms) of the neuron (Fig. 1c), when data from one autocorrelogram is fitted, or the cortical area when data from all neurons within that area are fitted together (Fig. 1d). Neurons from all areas, particularly ACC, showed evidence of lower correlation values at the shortest time lag[30]. This may reflect refractoriness or negative adaptation[31]. To overcome this, fitting started from the largest reduction in autocorrelation (between two consecutive time bins) onwards.

All recorded neurons were included in the population-level time-constant analysis (Fig. 1d). Single neurons were assigned a time-constant if their autocorrelogram could be reasonably described by an exponential decay[30]. Neurons were therefore automatically excluded if they had a fixation firing rate of <1 Hz or no decline in their autocorrelation function in the first 250 ms of time lags (27 of 694 excluded). Neurons were also excluded if the fitting produced extreme parameters ($A > 1.2$, $A < 0$, $\tau > 1000$, $\tau < 10$; 151 of 667 excluded). Finally, this was followed by a process of visual inspection where a further set of neurons were excluded which were considered to possess autocorrelation functions poorly characterised by an exponential decay (158 of 516 excluded). This left 136 DLPFC,

111 VLPFC, 73 OFC and 38 ACC neurons for analysis. Two independent observers completed this process, blind to each neuron's functional properties and recording location. The majority of excluded cells were recorded in ACC, where many neurons' autocorrelation functions were flat, possibly reflecting a timescale longer than could be indexed with a 1-s foreperiod. In VLPFC, which is the brain region where most analyses were performed, only 20.14% of all recorded neurons were excluded. All results were replicated without the visual inspection exclusion criteria.

**Decoding using linear-discriminant analysis**. A decoder based upon linear-discriminant analysis (LDA) was used to predict task features[20]. Decoding was performed separately for different task-types (i.e., SR or RS) and different task features (i.e., space and reward) in each neuronal population of interest. Neurons were pooled across sessions to create pseudopopulations. Neuronal firing rate was estimated at every 50 ms using a 100 ms window around the bin centre. This value was $z$-scored relative to the across-trial mean and standard deviation of firing rate in the final 300 ms of the fixation period. Decoders were built to classify either spatial location or reward size. Chance level performance for the classifier was therefore 12.5% (8 spatial locations) and 20% (5 reward levels), respectively. Half of all correct trials, grouped with a uniform distribution across conditions and reorganised into 1500 pseudotrials, were used as a training set. In the majority of analyses, the remaining correct trials were used to construct 100 pseudotrials to be used as a test set. The one exception was the error-trial analysis (Supplementary Figure 1), where error trials were used to construct the test set. Both sets of data were denoised using PCA at each timepoint. The data were reassembled using the minimal number of components sufficient to explain 90% of the variance. The purpose of this pre-processing step was to avoid singular matrices when LDA was performed and to reduce noise when using the decoder. The data were then input to the Matlab function *classify*. This process was repeated 1000 times to create a distribution of classifier performance.

Initially, the relevant task condition (reward or space) was predicted by training and testing the decoder on datasets from equivalent timepoints (Fig. 2a–d; Fig. 3). Classifier performance at a particular timepoint was determined to be significant if the 2.5th percentile of the distribution exceeded chance (Fig. 2a–d). To compare the classifier performance for high- and low-tau subpopulations (Fig. 3), the classifier performance for each subpopulation was averaged across 500 ms epoch windows within each permutation. The epochs were non-overlapping consecutive windows (0–500 ms of fixation; 500–1000 ms of fixation; 0–500 ms of first cue; 0–500 ms of delay-one; 500–1000 ms of delay-one; 0–500 ms of second cue; 0–500 ms of delay-two; 500–1000 ms of delay-two; 0–500 ms after go cue and 500 ms–1000 ms after go cue). For each epoch, a bootstrap test compared the distribution between the two populations[69]. Specifically, the population with the highest average performance across all permutations was determined (typically the high-tau population in epochs of interest). We then calculated the number of permutations where the value for the population with the lower average (typically the low-tau population) was above the value for the population with the higher average. The pairing of each permutation was arbitrary, so the pairings were randomly shuffled. This process was repeated 1000 times and the average was taken. The p value was this number divided by the total number of permutations. This was then corrected for multiple comparisons (Bonferroni correction for 10 epochs). This same bootstrap test was used to compare classifier performance when tested on correct trials and error trials (Supplementary Figure 1).

To further probe the temporal evolution of the neural coding, we then extended our approach so that for each timepoint a decoder was trained at, this decoder was also tested at all other points within the trial. Averaging performance across permutations created a timepoints x timepoints matrix of classifier performance (Supplementary Figure 2a). To investigate the effect of the second stimulus on the neural code, we defined 3 time periods as follows; T1 (the final 500 ms of delay-one), post-reward cue on (250–750 ms after reward cue onset), and T2 (the final 500 ms of delay-two). The performance of a decoder trained in T1 and tested in T1 (T1T1), trained in T2 and tested in T2 (T2T2), trained in T1 and tested in T2 (T1T2) and trained in T2 and tested in T1 were compared (Supplementary Figure 2e). Performance was averaged within the 500 ms time period for each permutation, then the difference between the two classifier performance distributions was compared to 0. If the 95th percentile range of the distribution did not overlap with 0, there was a significant difference in performance across time.

A final test was employed to verify if the weakening of the spatial code in SR-trials was caused by the presentation of the reward cue[20]. Supplementary Figure 2f shows the average performance (across permutations) of all classifiers trained in T1, tested at all timepoints. For each of the three epoch windows (T1, post-reward cue on, T2) a straight line was fitted to the data within each permutation. If the 95th percentile range of line gradients across permutations did not overlap with 0, there was a significant change in classifier performance in that window (Supplementary Figure 2g).

**Decoding using multivariate discriminability analysis**. To further explore the relationship between classifiers across time, and to highlight any inversions of tuning pattern across time, a correlation based classifier was used[35]. Tuning reversals would only appear weakly below chance using an accuracy-based classifier like in Fig. 2; therefore the correlation based discriminability analysis was more

suitable here. Decoding was performed separately for different task-types (i.e., SR or RS) and different task features (i.e., space and reward). For each neuron, correct trials were split equally into a training set and a test set. Within each set, trials were grouped according to the relevant feature to be decoded (either eight spatial groups or five reward levels). Neuronal firing rate was estimated at every 10 ms using a 200 ms window around the bin centre. Neuronal firing rate for each of the $c$ conditions was averaged across trials for each neuron producing a vector with length c. The pairwise difference (PWD) between neural firing in each of the conditions was calculated. For 8 spatial locations (5 reward levels) this produced 28 (10) PWDs. The Pearson's correlation-coefficient for each $PWD$ was calculated across neurons between the training set and the test set. These correlation coefficients were averaged using Fisher's Z-transformation to produce a single correlation-coefficient quantifying either reward discriminability or spatial discriminability. This process was repeated for each timepoint pair, so that the temporal profile of decodability could be evaluated (Fig. 4). In Fig. 4 the matrix of correlation coefficients was averaged across the diagonal in order for the data to reflect both training-to-test and test-to-training trial projections. In Fig. 6b, data from SR-trials was used as a training set, with data from RS-trials being used as a test set. A similar analysis was used to probe if the task being performed could be decoded (Fig. 6c). As above, trials were separated between the conditions to be decoded (RS-trials vs. SR-trials) within a training set and a test set. Neuronal firing rate for each of the two task types was averaged across trials for each neuron. The PWD of neural firing between task types was calculated. As there were only two task types, one PWD vector was produced for each set. This PWD was then correlated between the training and test sets across neurons at each individual timepoint.

Cluster-based permutation tests were used to correct for multiple comparisons while assessing the significance of time-series data[30,70]. Discriminability metrics were compared between the high and low-tau subpopulations using Fisher's-Z transformation (Supplementary Figure 5). This yielded a test-statistic at each timepoint pair. Test statistics were divided into a $10 \times 10$ grid of non-overlapping 500 ms × 500 ms windows beginning at fixation onset. Neighbouring pixels in each analysis window with an uncorrected (cluster-forming) threshold of $p < 0.05$ (one-tailed) were defined as candidate clusters. The size of the clusters were compared to a null distribution constructed using a permutation test. Neurons assigned to each subpopulation were randomly permuted 10,000 times and the cluster analysis was repeated for each permutation. The size of the largest cluster for each permutation was entered into the null distribution. The true cluster size was significant at the $p < 0.05$ ($p < 0.01$) level corrected if the true cluster length exceeded the 95th (99th) percentile of the null distribution. $p$ Values reported in the main text refer to the significance level for the largest cluster. A cluster's significance was determined to be $p < 0.0001$ if its length exceeded all those in the null distribution. A similar method was used to compare discriminability to chance levels (Figs. 4 and 6). Neighbouring pixels in each analysis window with an uncorrected (cluster-forming) threshold of $p < 0.01$ (two-tailed) were defined as candidate clusters. In this case, permuted clusters were calculated by shuffling the order of neurons in each of the PWDs in the test set.

**Across-trial correlation analysis**. Independent to selectivity measures, neural firing rate was correlated across the trial (Fig. 5a, b). Firing rate for each condition (eight spatial locations and five reward levels) was correlated across neurons between each timepoint pair. A separate training and test set were defined based upon a split half of the trials. The matrix of correlation coefficients plotted represents the average (using Fisher's Z-transformation) value across all of the conditions (Fig. 5a). For Fig. 5b, prior to performing the correlation, neural firing rate was demeaned within each condition at each timepoint for each neuron. Demeaning was performed at each timepoint, for each neuron; the mean firing rate across conditions was subtracted from the raw firing rate to generate the new value.

**State space analysis**. PCA was used to perform a state space analysis (Fig. 5, Supplementary Figure 6)[12]. Each subspace was defined using a training set of data averaged across half of the available trials for each neuron and tested using data from the remaining half. This makes stimulus-variance captured non-arbitrary (Fig. 5e) and explains why only a minimal amount of variance is explained in fixation before stimulus presentation. For each neuron, firing rate on training set trials was averaged for each condition for each timepoint. For the fixation and delay-one subspaces, activity was averaged across the relevant timepoints (fixation: −1000 to 0 ms relative to cue onset; delay-one: 500–1500 ms relative to cue onset). This produced a *Conds × Neurons* matrix. Activity was demeaned across conditions for each neuron. PCA was then performed over conditions to define a low-dimensional coding subspace for the two epochs within a high-dimensional neural state space. For the dynamic subspace, firing was not averaged across timepoints and the PCA was performed separately at each time point. Therefore, a slightly different subspace is produced for each time point. Once the principal components have been defined, we projected the left-out test set data onto the first two principal axes of the subspaces (Fig. 5c, d). The plotted traces therefore display a low-dimensional representation of the trajectory of population activity in the subspace across time.

To assess the generalisability of the delay-one subspace, we plotted the stimulus variance (SV) it captured across the trial relative to the fixation and dynamic

subspaces (Fig. 5e). At each timepoint $t$, SV was calculated using the following formula:

$$SV = \mathrm{Tr}\big(\mathbf{S}_k^T \mathbf{C}_t \mathbf{S}_k\big) \qquad (2)$$

In which $\mathbf{S}_k$ refers to the subspace defined from training data (limited to the first $k$ principal axes) and $\mathbf{C}_t$ refers to the across-stimuli covariance matrix of the test data at timepoint $t$ (see ref. 12. for further details). In our analyses, we used one fewer principal axes than the number of conditions (Space: $k = 7$; Reward: $k = 4$).

**Single neuron analyses.** For the preliminary single-neuron encoding analyses (Fig. 7a–h), a Kruskal–Wallis test was performed for spatial location and reward size at each time point. A cluster-based permutation test was performed to test for significance (Fig. 7e–h). Consecutive bins in each non-overlapping 500 ms analysis window with an uncorrected (cluster-forming) threshold of $p < 0.05$ were defined as candidate clusters. In this case, permuted clusters were calculated by shuffling the relevant feature (spatial location or reward size) across trials. This permutation was repeated for 1000 shuffles of the data. Each candidate cluster was significant at the $p < 0.05$ level (corrected) if its length exceeded the 95th percentile of the null distribution. To quantify the proportion of neurons showing stimulus selectivity throughout the trial, we split neural firing data into ten consecutive 500 ms epochs from fixation onset. We ran a separate Kruskal–Wallis test on the average firing rate of each Ventrolateral PFC neuron across these epochs. A subpopulation of neurons with selective responses during the initial cue presentation was defined ($n = 54$ for reward, $n = 62$ for space). The proportion of this subpopulation selective for each other factor was then calculated for all other epochs (Fig. 7i, j).

To probe whether neurons coding for both factors simultaneously demonstrated either linear or non-linear mixed selectivity, we performed a two-way ANOVA (Fig. 8).

**Smoothing time series figures.** Several graphs with time series data were smoothed across time bins for illustrative purposes (Figs. 6c; 7g, h; 8c, d). A moving average spanning five 10 ms bins was used. However, all statistical tests were performed on the unsmoothed data.

**Statistical methods.** The majority of our analyses made use of nonparametric permutation tests and, as such, did not make assumptions regarding the distribution of the data.

**Code availability.** All relevant code will be available from the corresponding authors on reasonable request.

**Data availability.** All relevant data will be available from the corresponding authors on reasonable request.

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

## Acknowledgements

S.E.C. was supported by the Middlesex Hospital Medical School General Charitable Trust. J.D.W. was supported by funding from NIMH R01-MH097990 and NIDA R21-DA035209. L.T.H. was supported by Henry Wellcome and Henry Dale Fellowships from the Wellcome Trust (098830/Z/12/Z and 208789/Z/17/Z), and by the NIHR Oxford Health Biomedical Research Centre. SWK was supported by NIMH (F32MH081521) and by a Wellcome Trust New Investigator Award (096689/Z/11/Z). We thank S. Mark for comments on an earlier draft of this manuscript. The views expressed are those of the authors and not necessarily those of the NHS, the NIHR or the Department of Health.

## Author contributions

J.D.W. and S.W.K. designed the task and collected the data. S.E.C., L.T.H. and S.W.K. conceived the analyses. S.E.C. and J.P.T. analysed the data. S.E.C., L.T.H. and S.W.K. wrote the manuscript.

## Additional information

**Competing interests:** The authors declare no competing interests.

