## [Peer Review File · Nature Communications]

Reviewers' comments:

Reviewer #1 (Remarks to the Author):

Cavanagh and colleagues present an analysis of neuronal activity in four regions of the PFC in a spatial oculomotor delayed response task with an intervening cue indicating reward size. They found that the intrinsic timescale (τ) of neuronal firing at rest varied across regions and was highest in ACC neurons, possibly following the neuronal processing hierarchy. Decoding analysis revealed that spatial information could be decoded best from VLPFC activity, and only in VLPFC spatial information was maintained after the reward-cue and prior to the response. In VLPFC, spatial selectivity was more sustained in high τ neurons than in low τ neurons, which could not be explained by stronger selectivity in general. Cross-temporal discriminability analysis of population activity in VLPFC high τ neurons showed that coding was dynamic at spatial cue presentation, stable during first delay and then distracted by the reward cue, indicating that the coding does not resemble a stable attractor. Low τ neurons exhibited dynamic coding with no generalization of the code across time. Population activity subspace analysis revealed that information was not stably represented from spatial cue onset on, but the code was reversed after spatial cue presentation and only stable during the delay. Cross-task discriminability analysis revealed stimulus-locked generalization of the spatial coding in VLPFC across the different trial types. A fraction of VLPFC neurons switched selectivity after presentation of the second cue. Some of these neurons show linear (not non-linear) mixed selectivity regarding the encoded information.

Overall, the paper addresses timely issues that are very relevant to the field of working memory coding. The results are presented in a clear and understandable fashion. However, in some important instances I feel that the conclusions the authors draw quite boldly are not well supported by the data.

Major points

First, the behavioral task used in this study has one important disadvantage: It is a delayed response task and not a “true” WM task, i.e. the behavioral response can be initiated already after presentation of the spatial cue. Therefore, from the presented data alone it cannot be assumed that the VLPFC sustained selectivity during the delay is behaviorally relevant and required for solving the task, since the response information could already have been relayed to a

downstream area and be maintained there. This is a possibility that the authors only briefly mention (line 471-473). This aspect limits the conclusions that can be drawn from the results. What we can conclude from the presented data is that spatial information is not represented by a stable attractor in PFC in a delayed response task with a “distractor” (see below for my comments on this term), which is at odds with what has been shown in a delayed response task and a true WM task without distractors (Murray et al., 2017). However, it is possible that in the present task the PFC network does not need to maintain the information and therefore can be easily distracted. Therefore, what we cannot say is whether WM is represented by a stable attractor in a true WM task, when the information needs to be maintained by a high executive. Maybe in this case, the PFC code would be stable even if a behaviorally relevant distractor is presented during the delay, while in the present task it simply does not need to be stable. The authors could be a little more modest with their claims and be more specific about the behavior (delayed response rather than true WM). I suggest that the authors show that the information in VLPFC is indeed behaviorally relevant. This could be implemented by an error trial analysis. Is the decoding performance during the delay and immediately prior to the response reduced in error trials? This could be shown, for example, using discriminability analysis (similar to Fig. 2/3 or 4) for correct trials vs. error trials.

Second, the authors repeatedly emphasize a particular strength of the task, namely that it contains a behaviorally relevant “distractor” (i.e. the reward cue). Indeed the reward cue is of behavioral relevance, especially since it impacts the animals’ motivation and activates a phylogenetically ancient brain system. However, this stimulus does not compete for the same resources as the spatial target, i.e. it is semantically different. The distinct semantics could explain why the results in this paper regarding e.g. linear vs. non-linear mixed selectivity diverge from previous research (the authors discuss Parthasarathy et al., 2017). It is conceivable that target and distractor information are not represented in a non-linearly mixed fashion if they are semantically far apart (Parthasarathy and colleagues found non-linear mixed selectivity for sample information and elapsed time, i.e. trial epoch information). Maybe even more convincingly, when target and distractor are semantically equivalent (Jacob & Nieder, 2014), there also seems to be non-linear mixed selectivity (interaction effect in Fig. S1 of Jacob & Nieder, 2014). This study also used a centrally presented distractor and could be mentioned in the discussion of this point (line 436/437). In sum, I would suggest not to over-emphasize the seeming contradictions to previous research, but to put the findings into perspective and highlight the differences in the behavioral tasks used.

Minor points

In order to allow other researchers to reproduce the presented work, a few more details are required:

- Fig 5a: Average of within-condition correlations
- Fig 6c, line 322, line 553: “Similar analysis”? What was calculated exactly?

Formal errors:

- Fig. 5 and 7: Axis labels are very small and difficult to see
- Fig. 7: X axis labels missing
- Fig.7: Caption for c) and d) cannot be found easily

Conclusion

The paper presents interesting analyses that further our understanding of how spatial information is maintained in prefrontal areas. With one additional piece of evidence (error trial analysis), and providing the conclusions are put into perspective and a few clarifications and changes in wording are made, I would be happy to support this paper being published in Nature Communications.

Reviewer #2 (Remarks to the Author):

In this article, Cavanaugh and colleagues examine the activity dynamics in populations of neurons across several areas of the frontal cortex during working memory. The study is a secondary analysis of data published previously (Kennerley et al., J. Neurophysiology 2009; Kennerley & Wallis, J. Neuroscience 2009). Neurophysiological data were obtained in these studies from monkeys trained to perform a challenging task varying the spatial location of targets to be remembered and the amount of reward to be expected (which turns out to be critical for the nature of neuronal encoding). The data are of high quality and the analysis of the data appropriate and interesting. The manuscript does many things well: it provides a lucid introduction of the field of working memory and its current debates and controversies. It replicates several findings of previous studies and provides a common framework for their interpretation. And it extends the analysis of how spatial and reward information is maintained in working memory in a more complex task than typically used in previous studies, providing novel insights. Some issues still need to be addressed in the current version of the manuscript.

1. The division of the lateral surface of the prefrontal cortex in dorsolateral and ventrolateral areas is questionable, as the authors include the lateral bank of the principal sulcus in the ventrolateral prefrontal cortex. Anatomical evidence suggests that both banks of the principal sulcus receive strong anatomical connections from the dorsal visual pathway e.g. Cavada and Goldman-Rakic, 1989. Lesions of the inferior convexity of the ventral prefrontal cortex produce very different types of deficits than lesions of the principal sulcus (Buckley et al., 2009). I

certainly sympathize with the authors for following the division scheme of their original publications but some of their conclusions seem questionable, namely that VLPC was the only region that discriminated between spatial locations and reward sizes; and that DLPFC had minimal ability to discriminate spatial location in the second delay period of the task, but also in the first delay period, and even during the cue presentation! Do the authors believe that every previous study of the DLPFC, in simpler but also more challenging tasks, that showed spatial encoding of visual stimuli was wrong? The authors may wish to either redefine the border of VLPFC beyond area 46v, or to pool DLPFC and VLPFC data for their analyses beyond figure 2 and steer clear of the anatomical division pitfall.

2. The paper sets up the importance of the time constant of the spike-count autocorrelation function for the stability of information in working memory. Yet, ACC, the area with the longest τ is not able to maintain information in memory at all. Perhaps instead of low and high τ neurons there is an optimal τ for working memory? Some further discussion of this finding is warranted.

3. Explain why and how the 24 spatial locations were collapsed to 8 spatial conditions. The main text refers to the Methods section, but that was not very informative, either.

4. Describe what quantity the Y axis of figure 2 represents in the figure legend or the text.

5. The finding of anti-correlation between cue and delay period activity was interesting, and the analysis in this context was novel. Such a pattern of activity has been described at the level of single neurons before (e.g. see reference 30).

6. Line 258: "Demeaning" may be misconstrued for "belittling". Rephrase.

7. The authors grapple with the question of where the spatial information is actually maintained after the task-relevant distractor appears in the middle of the trial and disrupts stable encoding. They end the paper suggesting that a stable mnemonic representation may be transmitted to oculomotor regions and only maintained in the form of a motor plan. But would that not be the frontal eye field in the case of the task being studied?

8. Typo in line 538 (second).

References:

Buckley, M.J., Mansouri, F.A., Hoda, H., Mahboubi, M., Browning, P.G., Kwok, S.C., Phillips, A., and Tanaka, K. (2009). Dissociable components of rule-guided behavior depend on distinct medial and prefrontal regions. *Science* 325, 52-58.

Cavada, C., and Goldman-Rakic, P.S. (1989). Posterior parietal cortex in rhesus monkey: II.

Evidence for segregated corticocortical networks linking sensory and limbic areas with the frontal lobe. *Journal of Comparative Neurology* 287, 422-445.

Reviewer #3 (Remarks to the Author):

The authors show how neurons in monkey PFC dynamically code spatial location and reward information. The description of previous literature is great, the results are generally well described as well, and the techniques and analysis methods seem sound. However, the results do not seem to contain a major new contribution, in particular relative to the recent papers by Jacob & Nieder 2014 and Parthasarathy et al. 2017 *Nature Neuroscience* (as also referenced in the manuscript).

The paper claims that there is a switching between the coding for spatial location and the reward, but this seems more like a general decline, in particular for SR trials. There is no comparison with trials without a second cue (the reward cue in SR trials and the location cue in the RS trials), which makes it hard to directly compare the influence of this second cue. Furthermore, no effort is made to quantify how abrupt the change was (as for example in Figure 2b and f in Parthasarathy et al. 2017 *Nature Neuroscience*).

The analysis of low versus high tau are novel, but the results are not that surprising. There is just a graded effect, neurons with long time constants show longer sustained activity than neurons with short time constants. As also mentioned in the manuscript, "it is not the case that low tau subpopulations are simply less task-selective" [line 186].

The comparison between areas is potentially interesting, but is not really exploited. The main effect that is found is that it's mostly the vIPFC that codes for spatial location. This is rather surprising with respect to previous literature which indicates that the dlPFC has a preference for spatial location while the vIPFC has a preference for object identity (e.g. Wilson, Scalaidhe & Goldman-Rakic, *Science* 1993; Riley, Qi & Constantinidis *Cerebral Cortex* 2016).

There was one result that I found interesting which is not really emphasized in the manuscript, switching seems to happen mostly for RS trials (e.g. Figure 2d), not SR trials (e.g. Figure 2a). So reward coding is lost when location cue comes up, but not that clearly in the other direction. This seems to suggest that spatial information is encoded more robustly than reward information. At least in the vIPFC this seems a clear result. (This also points to an interesting difference relative to the two previous papers on the subject, the two cues do not carry the same type of information, so the second cue is not simply 'distracting' information, it is additional information that is also relevant for the animal.) This makes it potentially interesting to look more close at ACC, as this area codes more robustly for the reward cue than the location cue. Unfortunately, the coding for location is so weak that it is hard to judge whether the switch happens in the SR trials in that case. At least for the ACC there seems to be no clear switch for RS trials (Figure 2d).

minor comment:

It might be helpful to represent cross-temporal plots with a white background, see https://github.com/kingjr/fix_your_jet

Text taken from the manuscript is within quotation marks and italicised. Any changes to the text are indicated with blue font.

Reviewers' comments:

Reviewer #1 (Remarks to the Author):

Cavanagh and colleagues present an analysis of neuronal activity in four regions of the PFC in a spatial oculomotor delayed response task with an intervening cue indicating reward size. They found that the intrinsic timescale (τ) of neuronal firing at rest varied across regions and was highest in ACC neurons, possibly following the neuronal processing hierarchy. Decoding analysis revealed that spatial information could be decoded best from VLPFC activity, and only in VLPFC spatial information was maintained after the reward-cue and prior to the response. In VLPFC, spatial selectivity was more sustained in high τ neurons than in low τ neurons, which could not be explained by stronger selectivity in general. Cross-temporal discriminability analysis of population activity in VLPFC high τ neurons showed that coding was dynamic at spatial cue presentation, stable during first delay and then distracted by the reward cue, indicating that the coding does not resemble a stable attractor. Low τ neurons exhibited dynamic coding with no generalization of the code across time. Population activity subspace analysis revealed that information was not stably represented from spatial cue onset on, but the code was reversed after spatial cue presentation and only stable during the delay. Cross-task discriminability analysis revealed stimulus-locked generalization of the spatial coding in VLPFC across the different trial types. A fraction of VLPFC neurons switched selectivity after presentation of the second cue. Some of these neurons show linear (not non-linear) mixed selectivity regarding the encoded information.

Overall, the paper addresses timely issues that are very relevant to the field of working memory coding. The results are presented in a clear and understandable fashion. However, in some important instances I feel that the conclusions the authors draw quite boldly are not well supported by the data.

Thank you for a detailed summary of our findings and acknowledging the paper addresses timely issues in the working memory field. The reviewer raises several important points, and we will address each of these below.

Major points

First, the behavioral task used in this study has one important disadvantage: It is a delayed response task and not a "true" WM task, i.e. the behavioral response can be initiated already after presentation of the spatial cue. Therefore, from the presented data alone it cannot be assumed that the VLPFC sustained selectivity during the delay is behaviorally relevant and required for solving the task, since the response information could already have been relayed to a downstream area and be maintained there. This is a possibility that the authors only briefly mention (line 471-473). This aspect limits the conclusions that can be drawn from the results. What we can conclude from the presented data is that spatial information is not represented by

a stable attractor in PFC in a delayed response task with a “distractor” (see below for my comments on this term), which is at odds with what has been shown in a delayed response task and a true WM task without distractors (Murray et al., 2017). However, it is possible that in the present task the PFC network does not need to maintain the information and therefore can be easily distracted. Therefore, what we cannot say is whether WM is represented by a stable attractor in a true WM task, when the information needs to be maintained by a high executive. Maybe in this case, the PFC code would be stable even if a behaviorally relevant distractor is presented during the delay, while in the present task it simply does not need to be stable. The authors could be a little more modest with their claims and be more specific about the behavior (delayed response rather than true WM).

We thank the reviewer for raising this important point. Our task is indeed an oculomotor delayed response task, and we appreciate the reviewers’ distinction about a “True WM” task (Jacob & Nieder, 2014; Lundqvist et al., 2016; Mendoza-Halliday & Martinez-Trujillo, 2017; Meyers, Freedman, Kreiman, Miller, & Poggio, 2008; Qi et al., 2010; Rainer & Miller, 2002; Romo, Brody, Hernandez, & Lemus, 1999; Stokes et al., 2013; Warden & Miller, 2010) from tasks (like ours) where the mnemonic information is correlated with the required response. We have therefore sought to clarify this issue by making the following changes to the manuscript:

In the abstract we now say:

“Competing accounts propose that working memory (WM) is subserved either by persistent activity in single neurons or by dynamic (time-varying) activity across a neural population. Here we compare these hypotheses across four regions of prefrontal cortex (PFC) in an oculomotor-delayed-response task, where an intervening distractor indicated the reward available for a correct saccade.”

In the introduction we now say:

“We tested these hypotheses in an oculomotor-delayed-response task where a stimulus revealing the reward for a correct response was presented either before or after the spatial cue which the subject had to maintain in WM.”

In the results we now say:

*“Two rhesus macaques (*Macaca mulatta*) performed an oculomotor-delayed-response task requiring spatial WM, where the reward amount for successful responses varied across trials (**Fig. 1a**) (Kennerley & Wallis, 2009a, 2009b).”*

In the discussion we now say:

“Here we used an oculomotor-delayed-response task with a distracting reward cue to test whether working memory (WM) is subserved by persistent activity in single neurons or by dynamic activity across a neural population.”

In the discussion we now say:

“By probing the effect of a salient reward cue on the stability of mnemonic representations, we were able to further test whether cortical attractors in PFC provide a mechanism for distractor-resistant WM during an oculomotor-delayed-response task. It was shown that the intervening reward cue quenched the WM selectivity pattern in the VLPFC population.”

In the discussion we now say:

“An important question therefore remains how WM is achieved on this task. The residual level of dynamic VLPFC spatial coding may be sufficient for WM performance. Another possibility, although not directly verifiable with our data, is that a PFC region maintains a representation of the mnemonic stimulus in an activity silent state (Mongillo, Barak, & Tsodyks, 2008; Stokes, 2015). Alternatively, PFC may be essential for setting up a stable mnemonic representation during the initial delay, but if this mnemonic information specifies a response that could be prepared, this information could then be transmitted to oculomotor regions, such as frontal eye field or superior colliculus, to prepare a saccade. This would be akin to activity in premotor regions for reaching movements (Cisek, 2007). It would be worthwhile to explore these ideas in a task where the contents of WM were independent of the response (Rainer & Miller, 2002; Romo et al., 1999; Warden & Miller, 2010), and investigate the impact of a salient distractor on the WM representation. Regardless, our data are incompatible with PFC maintaining WM in cortical attractor networks throughout an oculomotor-delayed-response task, when the delay is interrupted with a behaviourally relevant distractor. This provides novel neurophysiological evidence that stable activity states within PFC may be more tightly associated with the most-recently presented behaviourally-relevant stimulus, rather than the contents of WM.”

In the Figure 1 legend we now say:

“Figure 1: Overview of reward-varying oculomotor-delayed-response task, recording locations and time constant analysis. a) Reward-varying oculomotor-delayed-response task. Monkeys were trained to remember a spatial position in working memory. They were also presented with a cue indicating the reward size they would receive for successfully completing the trial with a saccade to the remembered location. On RS (Reward-Space) trials, the reward cue was presented first; whereas on the SR (Space-Reward) trials, the cues were presented in the reverse order. On SR trials the reward cue therefore acted as a distraction to working memory of the task-relevant spatial information.”

However, we would also like to emphasise a few important points which suggest our findings may not be limited to delayed response tasks. Firstly, Murray and colleagues find largely similar working memory correlates within lateral prefrontal cortex (LPFC) for the oculomotor delayed response (ODR) task and the vibrotactile delayed discrimination (VDD) task (Murray, Bernacchia, et al., 2017). Furthermore, the pattern of population activity we observe (Fig4) up until the end of delay one (before the distractor) is also consistent with what was previously reported for the ODR and VDD tasks (Murray, Bernacchia, et al., 2017). While the required action following the delay is different for the two tasks (ODR – report remembered location; VDD – discriminate between a second stimulus), the literature suggests LPFC maintains the necessary information in WM using a similar pattern of population activity. Thus, although exploring the effect of a behaviourally relevant distractor during a “True WM” task would extend our findings, if VLPFC encodes the contents of current attention as a mnemonic representation, it is plausible that similar results would be observed. Finally, despite the fact that our task cannot attest to the possibility that PFC attractor dynamics would be resistant to distractors on a “True WM” task, demonstrating that PFC WM attractor dynamics are disrupted during delayed response task remains a significant challenge to the latest circuit models of WM (Murray, Bernacchia, et al., 2017; Murray, Jaramillo, & Wang, 2017). These models still maintain that PFC attractors underlie WM on delayed response tasks.

I suggest that the authors show that the information in VLPFC is indeed behaviorally relevant. This could be implemented by an error trial analysis. Is the decoding performance during the delay and immediately prior to the response reduced in error trials? This could be shown, for example, using discriminability analysis (similar to Fig. 2/3 or 4) for correct trials vs. error trials.

Following the reviewer's suggestion, we also performed a decoding analysis comparing correct and error trials (Parthasarathy et al. 2017). This showed that at certain points of a trial the remembered spatial location was better decoded from VLPFC activity on correct trials than error trials. These points came during initial stimulus presentation, but also during the initial delay and around the time of response. This provides strong evidence that VLPFC activity plays an important role in encoding and then maintaining spatial working memory during this task. We have included this as an additional supplementary figure, and thank the reviewer for this important suggestion.

Supplementary Figure 1. Error trial analysis shows VLPFC spatial activity on SR trials is behaviourally relevant. a) Mean performance (across 1000 permutations) of spatial decoding through the trial. The decoder was trained on correct trials and tested on data from left-out correct trials (dark blue) or error trials (lighter blue). Dashed line shows chance-level performance. **b)** Comparison of classifier performance across the trial. Boxplots show the distribution of classifier accuracies within each epoch for each permutation. Each epoch has a pair of boxplots; the left-side for correct trials (dark blue) and the right-side for error trials (lighter blue). The area contained within the whiskers of the boxplots represents the 95th percentile range of classifier performance. The central mark is the median of the distribution. Performance for correct and error trials was

compared within each 500ms epoch using a Bonferroni-corrected bootstrap test (see Methods). On correct trials, the decoding accuracy is significantly higher during stimulus presentation, the initial delay, and around the time of response (***, $p < 0.001$).

It is now referenced in the results section as follows:

“Our results provide the most complete comparison to date of population-level WM activity patterns across multiple PFC brain regions (Fig.2). Of the four PFC regions examined, VLPFC activity best discriminated between both the different spatial locations and the different reward sizes regardless of trial (SR, RS) type, and it was the only PFC region that sustained both of these selectivity patterns across delays. VLPFC was also the PFC region most strongly discriminating spatial information immediately prior to saccade. To examine whether VLPFC activity was behaviourally relevant, we performed an error-trial analysis. We found that decoding accuracy was significantly stronger on correct SR trials relative to error trials, and this effect was evident during spatial cue presentation, the initial WM delay, and during saccade preparation (Supplementary Fig.1).”

Second, the authors repeatedly emphasize a particular strength of the task, namely that it contains a behaviorally relevant “distractor” (i.e. the reward cue). Indeed the reward cue is of behavioral relevance, especially since it impacts the animals’ motivation and activates a phylogenetically ancient brain system. However, this stimulus does not compete for the same resources as the spatial target, i.e. it is semantically different. The distinct semantics could explain why the results in this paper regarding e.g. linear vs. non-linear mixed selectivity diverge from previous research (the authors discuss Parthasarathy et al., 2017). It is conceivable that target and distractor information are not represented in a non-linearly mixed fashion if they are semantically far apart (Parthasarathy and colleagues found non-linear mixed selectivity for sample information and elapsed time, i.e. trial epoch information). Maybe even more convincingly, when target and distractor are semantically equivalent (Jacob & Nieder, 2014), there also seems to be non-linear mixed selectivity (interaction effect in Fig. S1 of Jacob & Nieder, 2014). This study also used a centrally presented distractor and could be mentioned in the discussion of this point (line 436/437). In sum, I would suggest not to over-emphasize the seeming contradictions to previous research, but to put the findings into perspective and highlight the differences in the behavioral tasks used.

We are very grateful to the reviewer for this excellent point; this seems an interesting way of interpreting our mixed selectivity results with respect to the previous literature.

In the discussion we now say:

“The dynamic switch of VLPFC activity to coding the behaviourally relevant distractor provides further evidence that PFC neurons can be tuned to multiple diverse cognitive factors, and that they can flip between them within the course of a trial (Blackman et al., 2016; Enel, Procyk, Quilodran, & Dominey, 2016; Rigotti et al., 2013; Spaak, Watanabe, Funahashi, & Stokes, 2017). It also suggests previous studies concluding PFC neurons are resistant to distraction do not generalise to more behaviourally salient stimuli (Lennert & Martinez-Trujillo, 2011; Qi et al., 2010; Suzuki & Gottlieb, 2013). Here we used a reward-predictive cue presented at the fixation spot, as opposed to a peripherally flashed target (Suzuki & Gottlieb, 2013) or stimulus (Lennert & Martinez-Trujillo, 2011; Aishwarya Parthasarathy et al., 2017) which is irrelevant to the task. Our results concur with those of

Jacob and Nieder (2014), who also observed distractor encoding within PFC when a centrally presented (but currently task-irrelevant) stimulus interrupted a WM delay.

The flexibility with which VLPFC neurons changed the factor they encoded also has implications for accounts of mixed selectivity (Aishwarya Parthasarathy et al., 2017; Raposo, Kaufman, & Churchland, 2014; Rigotti et al., 2013). Shortly after the second stimulus was shown, there was evidence for neurons encoding a combination of factors. However, we found the majority of this mixed selectivity was linear (Raposo et al., 2014), as opposed to non-linear (Jacob & Nieder, 2014; Aishwarya Parthasarathy et al., 2017; Rigotti et al., 2013). This may reflect a difference between the behavioural tasks used across studies. Non-linear mixed selectivity has predominantly been observed for task features which are semantically similar (Jacob & Nieder, 2014; Lebedev, Messinger, Kralik, & Wise, 2004; Aishwarya Parthasarathy et al., 2017; Rigotti et al., 2013; Spaak et al., 2017), rather than those which are far apart (Raposo et al., 2014). It may be that PFC neurons did not encode an interaction of spatial location and reward size because they are semantically distinct.”

Minor points

In order to allow other researchers to reproduce the presented work, a few more details are required:

- Fig 5a: Average of within-condition correlations

Thank you to the reviewer for raising this point. We hope the following makes this clearer:

The main text in the original submission said:

“To examine this issue in more detail, we correlated activity within the VLPFC high tau subpopulation across time within each condition (Fig5a, see Methods)”

The methods in the original submission said:

“Independent to selectivity measures, neural firing rate was correlated across the trial (Fig5a, b). Firing rate for each condition (eight spatial locations, five reward levels) was correlated across neurons between each timepoint pair. A separate training and test set were defined based upon a split half of the trials. The matrix of correlation coefficients plotted represents the average (using Fisher’s Z-transform) value across all of the conditions (Fig5a). For Fig5b, prior to performing the correlation, neural firing rate was demeaned within each condition and timepoint for each neuron.”

We believe the confusion may have arisen because of the Figure 5 legend. We have updated it accordingly:

“a) Within-condition correlation (see Methods) of neural firing across time for SR trials. All bins are positively correlated with each other, suggesting neural firing is stable across time. Note positive correlation between cue period and delay (asterisk).”

- Fig 6c, line 322, line 553: “Similar analysis”? What was calculated exactly?

Thank you for raising this point. We have added the following detail to the methods section:

*“A similar analysis was used to probe if the task being performed could be decoded (**Fig.6c**). As above, trials were separated between the conditions to be decoded (RS trials vs. SR trials) within a training set and a test set. Neuronal firing rate for each of the two task types was averaged across trials for each neuron. The PWD of neural firing between task types was calculated. As there were only two task types, one PWD vector was produced for each set. This PWD was then correlated between the training and test sets across neurons at each individual timepoint.”*

Formal errors:

- Fig. 5 and 7: Axis labels are very small and difficult to see
- Fig. 7: X axis labels missing
- Fig.7: Caption for c) and d) cannot be found easily

Thank you for these helpful points. These have been addressed.

Conclusion

The paper presents interesting analyses that further our understanding of how spatial information is maintained in prefrontal areas. With one additional piece of evidence (error trial analysis), and providing the conclusions are put into perspective and a few clarifications and changes in wording are made, I would be happy to support this paper being published in Nature Communications.

Thank you for a well-structured review with excellent suggestions for how we could improve the manuscript. We hope that the error trial analysis has provided additional evidence that the activity of VLPFC is important for task performance. We have tried to put our conclusions into perspective with regards to specifically referring to the task as an oculomotor delayed response task, and noting that demonstrating the effects of a behaviourally-relevant distractor on WM representations on a “True WM” task would extend our conclusions. We took on board the reviewer’s point about the role of the behavioural task in non-linear mixed selectivity and incorporated this into our discussion.

Reviewer #2 (Remarks to the Author):

In this article, Cavanaugh and colleagues examine the activity dynamics in populations of neurons across several areas of the frontal cortex during working memory. The study is a secondary analysis of data published previously (Kennerley et al., J. Neurophysiology 2009; Kennerley & Wallis, J. Neuroscience 2009). Neurophysiological data were obtained in these studies from monkeys trained to perform a challenging task varying the spatial location of targets to be remembered and the amount of reward to be expected (which turns out to be critical for the nature of neuronal encoding). The data are of high quality and the analysis of the data appropriate and interesting. The manuscript does many things well: it provides a lucid introduction of the field of working memory and its current debates and controversies. It

replicates several findings of previous studies and provides a common framework for their interpretation. And it extends the analysis of how spatial and reward information is maintained in working memory in a more complex task than typically used in previous studies, providing novel insights. Some issues still need to be addressed in the current version of the manuscript.

We thank the reviewer for acknowledging the interesting nature of our findings and providing suggestions which we feel have improved the manuscript.

1. The division of the lateral surface of the prefrontal cortex in dorsolateral and ventrolateral areas is questionable, as the authors include the lateral bank of the principal sulcus in the ventrolateral prefrontal cortex. Anatomical evidence suggests that both banks of the principal sulcus receive strong anatomical connections from the dorsal visual pathway e.g. Cavada and Goldman-Rakic, 1989. Lesions of the inferior convexity of the ventral prefrontal cortex produce very different types of deficits than lesions of the principal sulcus (Buckley et al., 2009). I certainly sympathize with the authors for following the division scheme of their original publications but some of their conclusions seem questionable, namely that VLPFC was the only region that discriminated between spatial locations and reward sizes; and that DLPFC had minimal ability to discriminate spatial location in the second delay period of the task, but also in the first delay period, and even during the cue presentation! Do the authors believe that every previous study of the DLPFC, in simpler but also more challenging tasks, that showed spatial encoding of visual stimuli was wrong? The authors may wish to either redefine the border of VLPFC beyond area 46v, or to pool DLPFC and VLPFC data for their analyses beyond figure 2 and steer clear of the anatomical division pitfall.

We thank the reviewer for bringing up these important points about lateral prefrontal cortex anatomy. We hope to address three points separately:

- 1) Why does ventrolateral PFC (VLPFC) have stronger working memory selectivity than dorsolateral PFC (DLPFC)
- 2) Redefining the border of VLPFC to an alternative definition
- 3) Why does DLPFC have minimal ability to discriminate spatial location

1) Why does VLPFC have stronger working memory selectivity than DLPFC

We agree with the reviewer that DLPFC, and particularly within principal sulcus (PS) in macaque monkeys, is the brain region most commonly associated with spatial working memory. However, although often overlooked, even some of the seminal electrophysiology studies on oculomotor delayed response (Funahashi, Bruce, & Goldman-Rakic, 1989, 1990, 1991) show spatially-sensitive neurons recorded ventral to the PS. More recent studies have demonstrated working memory correlates within VLPFC when spatial (Hoshi, Shima, & Tanji, 2000; Lebedev et al., 2004; Rainer, Asaad, & Miller, 1998; Rao, Rainer, & Miller, 1997) and other features of stimuli (Freedman, Riesenhuber, Poggio, & Miller, 2003; Jacob & Nieder, 2014; Rainer et al., 1998; Rainer & Miller, 2002; Rao et al., 1997; Romo et al., 1999) must be remembered. As we discuss further below, studies that require subjects to attend to spatial information *in addition* to other task features (e.g., reward information, object identity, task rule), as in the current task, demonstrate particularly strong WM selectivity in VLPFC (Hoshi et al., 2000; Kennerley & Wallis,

2009b; Lebedev et al., 2004; Rao et al., 1997). Moreover, in studies where a sufficiently large sample of neurons were recorded in both DLPFC and VLPFC to allow quantitative comparisons, a greater proportion of neurons in VLPFC have spatially tuned responses (Hoshi et al., 2000; Kennerley & Wallis, 2009b; Lebedev et al., 2004). Thus, although we agree that traditionally DLPFC has been the brain area more often associated with spatial working memory, there are a number of studies that corroborate our finding of stronger spatial selectivity within VLPFC. However, we prefer to keep the focus of this paper on the relationship between neuronal time constant and WM-related activity, and let the data speak for themselves (see below).

2) Redefining the border of VLPFC to an alternative definition

We acknowledge the reviewer’s point that our demarcation of VLPFC and DLPFC was slightly controversial. In the original Kennerley et al. 2009 paper, we distinguished between DLPFC/VLPFC at the fundus of the Principal Sulcus (PS) because i) anatomical evidence suggests the dorsal and ventral banks of PS receive different connections (Cavada & Goldman-Rakic, 1989; Petrides & Pandya, 1984) and ii) the Paxinos rhesus monkey brain atlas (Paxinos, Huang, & Toga, 2000) and the Petrides PFC nomenclature (i.e.,(Petrides & Pandya, 1999)) both use “ventral” to describe areas within and around the ventral bank of PS (i.e., 46v, 9/46v); it seemed confusing to include “ventral” areas within our classification of DLPFC, though we acknowledge that both banks of PS have traditionally been part of “DLPFC”. However, recent anatomical work suggests the fundus of PS divides the “VLPFC” and “DPFC” networks (Saleem, Miller, & Price, 2014), so our original division of DLPFC/VLPFC was not without merit.

To explore whether the precise boundary of DLPFC/VLPFC influenced our results, we have examined the recording location of every LPFC neuron and given it an area coding based on the Paxinos rhesus monkey brain atlas (Paxinos et al., 2000). In our original submission, this led to neurons categorized as follows:

DLPFC	VLPFC
46d, 9/46d within Principal Sulcus	46v, 9/46v within Principal Sulcus
9/46d outside Principal Sulcus	9/46v outside Principal Sulcus
8AD	8AV
8B	45A and 45B
9	47/12

We investigated how the results presented in Figure 2 were impacted by redefining the boundaries of these two brain regions. We have compared the selectivity patterns of DLPFC neurons within and around the dorsal bank of PS against the selectivity patterns of DLPFC neurons within and around the ventral bank of PS as a new supplementary figure. Note that in our new definition of DLPFC, we excluded neurons ~2-10mm dorsal to the principal sulcus in areas 8AD, 8B, 9L as these aren’t typically considered part of DLPFC.

Supplementary Figure 3: Decoding performance of different populations of dorsolateral prefrontal cortex (DLPFC) and ventrolateral prefrontal cortex (VLPFC) neurons. In the main paper, we defined DLPFC as including all neurons within and around the dorsal bank of the Principal Sulcus (PS) in areas 9/46D and 46D (black line) and all neurons within and around the ventral bank of the PS in areas 46V and 9/46V (orange line). Note both 9/46D and 9/46V extend outside of the dorsal and ventral banks, respectively, by approximately 2mm. The Paxinos et al 2000 macaque monkey brain atlas was used to define the brain area of each recorded neuron. The mean performance of a classifier (1000 permutations, see **Methods**) trained to decode each task feature (spatial location, **a** and **c**; reward level, **b** and **d**) are plotted for each neural population and trial type (SR-task, **a** and **b**; RS-task, **c** and **d**). The first solid vertical line signifies when subjects were cued to respond. The first and second dashed vertical lines represent the average timing of the subjects' saccade and the onset of reward respectively. Solid coloured horizontal lines represent significant encoding for the corresponding brain region (2.5th percentile of distribution > chance level, $p < 0.05$). The dashed magenta line represents chance level classifier performance.

This figure clearly demonstrates that:

- i) Extending the definition of “DLPFC” to include areas 46v and 9/46v within the ventral bank of PS does not increase the selectivity patterns of DLPFC
- ii) The selectivity patterns in all neural populations including and dorsal to 9/46v are relatively weaker than the spatial selectivity observed in the inferior convexity.
- iii) Regardless of the anatomical division chosen, our data show stronger spatial selectivity within VLPFC than DLPFC.

We have now included an additional note within the main results section of the manuscript:

“Maintenance of spatial discriminability in DLPFC was comparatively weaker, emerging relatively late in the spatial cue epoch and decaying shortly after the first delay (Fig.2a, c). This is surprising given that DLPFC has often been implicated in the stable maintenance of WM, but such discrepancies may be due to variability in recordings along the anterior-posterior gradient of DLPFC (Riley, Qi, & Constantinidis, 2016), or studies describing DLPFC

cells or lesions which extend to surrounding areas including VLPFC(Bauer & Fuster, 1976; Funahashi et al., 1989; Funahashi, Bruce, & Goldman-Rakic, 1993). Further analysis revealed that our results were not dependent upon whether the border between VLPFC and DLPFC was within or just ventral to the principal sulcus (Supplementary Fig.3)."

However, we also agree with the reviewer that adopting a more conventional categorisation of DLPFC, by including neurons both within, and just outside (~0-2mm) the banks of the principal sulcus (PS) would help comparison across studies. We have thus redefined the brain areas as:

DLPFC	VLPFC
9/46d ~0-2mm dorsal to Principal Sulcus	8AV
9/46d, 46d, 46v, 9/46v within Principal Sulcus	45A and 45B
9/46v ~0-2mm ventral to Principal Sulcus	47/12

We have now reanalysed all the data and reproduced all of the figures according to the above classification schema. We have updated the Methods with new definitions for VLPFC and DLPFC:

"Single neurons were recorded from four brain regions across prefrontal cortex (PFC; Fig.1b): anterior cingulate cortex (ACC; areas 9m, 24c, n= 198), dorsolateral PFC (DLPFC; areas 9/46d, 46d, 46v, 9/46v n= 205), ventrolateral PFC (VLPFC; areas 8AV, 45A, 45B, 47/12, n= 139) and orbitofrontal cortex (OFC; areas 11, 13, n= 152). The brain areas associated with each recorded neuron were estimated using the macaque monkey brain atlas(Paxinos et al., 2000), following physiological validation of anatomical landmarks (i.e., sulci) to reconstruct the precise locations of all recorded neurons(Kennerley & Wallis, 2009a, 2009b). For this report, we classified all neurons recorded inside the principal sulcus (PS), and within ~2mm of lateral surface both dorsal and ventral to the PS (in areas 9/46d and 9/46v), as DLPFC. All neurons in brain areas classified as VLPFC were thus located ~>2mm ventral to the PS.

3) Why does DLPFC have minimal ability to discriminate spatial location

We agree with the reviewer that the low level of spatial coding within DLPFC is surprising. Indeed, we did originally mention in the results section:

"This is surprising given that DLPFC has often been implicated in the stable maintenance of working memory, but such discrepancies may be due to variability in recordings along the anterior-posterior gradient of DLPFC, or studies describing DLPFC cells or lesions which extend to surrounding areas including VLPFC."

However, it is important to keep in mind that despite the fact we find spatial selectivity is stronger in VLPFC than DLPFC, DLPFC neurons still encode spatial information. Our original discriminability analysis was a stringent test of spatial coding. Even so, there was significant coding of the spatial location during both tasks while the stimulus is visible (old Fig2a, Fig2C). This spatial coding also remained above baseline for the majority of the subsequent delay (Delay 1 SR Trials; Delay 2 RS Trials), although not statistically significant. In order to help us address the concerns of reviewer 3, we have now switched to a 'decoding' method based upon linear-discriminant analysis (A. Parthasarathy et al., 2017). This analysis shows more clearly a

prolonged, above chance level coding for spatial location and reward size in DLPFC (new Figure 2, see below).

Figure 2: Ventrolateral prefrontal neurons maintain information for both spatial and reward stimuli during delay epochs. The mean performance of classifiers (1000 permutations, see Methods) trained to decode each task feature (spatial location, **a** and **c**; reward level, **b** and **d**) are plotted for each brain area and trial type (SR-task, **a** and **b**; RS-task, **c** and **d**). Ventrolateral prefrontal cortex (VLPFC) is the only region to strongly code information about space and reward across the trial. Notably, the VLPFC activity primarily encodes the factor most recently presented. When the reward cue is shown first (RS task, **c** and **d**), a representation of reward size is maintained throughout the first delay, but falls away when the spatial cue is presented. More surprisingly, a similar weakening of spatial coding is also observed on the SR Task (**a**), even though this analysis is restricted to trials where the subject remembered the correct spatial location. The VLPFC population strongly encodes and maintains a representation of the remembered spatial location, but this is substantially weakened by the offset of the reward cue. The first solid vertical line signifies when subjects were cued to respond. The first and second dashed vertical lines represent the average timing of the subjects' saccade and the onset of reward respectively. Solid coloured horizontal lines represent significant encoding for the corresponding brain region (2.5^{th} percentile of distribution > chance level, $p < 0.05$). The dashed magenta line represents chance level classifier performance.

Importantly, the results are also consistent with what we found in our previous papers using a single-neuron encoding analysis (compare (Kennerley & Wallis, 2009b) **Fig5D** with **Fig2A** in our submission). We would also like to point out that we do not attempt to make a strong claim about DLPFC activity in our manuscript; and are not dismissive of the previous literature which has observed stronger spatial selectivity in this region. Rather, we quickly isolate VLPFC as the area showing the strongest working memory correlates and focus upon that region for the remainder of the paper.

We have extended our discussion of the DLPFC results and provided some potential explanations of our findings, relative to the previous literature:

“Of the PFC regions studied, VLPFC mnemonic representations were the strongest, and the only ones present during the second delay of SR trials, although in an altered state relative to the initial delay. DLPFC had relatively weaker and short-lived coding of spatial location

(although more prolonged than ACC or OFC). This is perhaps surprising, as some hypotheses propose DLPFC primarily maintains WM for spatial locations, whereas VLPFC maintains WM for object identities(Riley et al., 2016; Wilson, Scialoja, & Goldman-Rakic, 1993). However, strong connections exist between parietal cortex and VLPFC(Cavada & Goldman-Rakic, 1989; Petrides & Pandya, 1984) which could provide spatial information, and VLPFC receives input from high-level visual areas in the temporal lobe(Barbas, 1988; Petrides & Pandya, 1999) which could provide object-related information. Perhaps consequently, other reports, including our previous single neuron analyses of this dataset(Kennerley & Wallis, 2009b), observe equivalent or stronger spatial selectivity ventral to the principal sulcus(Hoshi et al., 2000; Lebedev et al., 2004; Rao et al., 1997). These studies all required subjects to attend to spatial information in addition to other task features (e.g., reward information, object identity, task rule). Thus, it is possible that spatial selectivity may shift toward VLPFC in contexts where flexible allocation of attention to multiple task features is required. Indeed, the flexible shifting of VLPFC selectivity that we observed (Fig.7) is consistent with the idea that VLPFC encodes the focus of attention(Lebedev et al., 2004)."

2. The paper sets up the importance of the time constant of the spike-count autocorrelation function for the stability of information in working memory. Yet, ACC, the area with the longest tau is not able to maintain information in memory at all. Perhaps instead of low and high tau neurons there is an optimal tau for working memory? Some further discussion of this finding is warranted.

We thank the reviewer for this helpful point. In both the current report and our previous time constant publication, ACC neurons strongly encoded reward information at the cue, but this was not strongly maintained in the delay (Figure 2; see Figure 4 of (Cavanagh, Wallis, Kennerley, & Hunt, 2016)). However, whilst high tau cells might support working memory functions, a diversity of time constants might also support other cognitive processes which operate over different timescales, such as in learning and decision-making; two functions often associated with ACC. Indeed a diversity of learning rates have recently been reported in ACC (Meder et al., 2017), and ACC neurons exhibit a diversity of timescales for reward integration (Bernacchia, Seo, Lee, & Wang, 2011). How tau relates to ACC function is an area of current research in our lab. We have amended our discussion section as follows:

"Both attractor(Brunel & Wang, 2001; Murray, Bernacchia, et al., 2017; Wang, 1999) and synaptic models(Mongillo et al., 2008) of WM stress the importance of a recurrent network architecture. By using the decay of autocorrelation of spiking activity during a fixation period as an unbiased metric of intrinsic persistent activity, we demonstrate that neurons with higher time constants (taus) are more likely to exhibit WM-related selectivity, but only in the VLPFC population. The VLPFC high-tau subpopulation had stable selectivity during the initial delay period following stimulus offset, whereas the low-tau subpopulation exhibited dynamic coding. Importantly, any distinction between the high and low-tau VLPFC subpopulations was only evident during this mnemonic phase, ruling out the possibility that high-tau cells are simply more task-selective. However, ACC, the PFC region where the longest taus were observed, did not display prolonged selectivity for either reward or spatial information. This may suggest that there is an optimal tau required to support WM. However, an alternate explanation is that a neuron's selectivity pattern is constrained by both the functional anatomy of its brain region and its tau. Long tau neurons in ACC may perform complex

functions across longer timescales than our task was designed to investigate, such as the integration of information across trials(Bernacchia et al., 2011)."

3. Explain why and how the 24 spatial locations were collapsed to 8 spatial conditions. The main text refers to the Methods section, but that was not very informative, either.

We agree this was ambiguous. We have shown how the conditions were collapsed into 8 triangles in a small inset of Figure 5C. We collapsed the 24 conditions into 8 conditions to allow for sufficient trials for the decoding analyses; we had to further collapse the trials into 4 conditions for sufficient power for the error analysis for Reviewer 3. We hope this clarifies how and why we collapsed across conditions. This is now referenced in the methods section as follows.

"The 24 spatial targets were collapsed into eight locations forming triangles (Fig.5C, inset) to allow for sufficient trials for the decoding analyses. For the error trial analysis (Supplementary Fig.1), the 24 spatial targets were collapsed into four locations by combining tessellating pairs of triangles into rectangles."

4. Describe what quantity the Y axis of figure 2 represents in the figure legend or the text.

We have altered Figure 2 to a different style of analysis for consistency given Reviewer 3's comments. Hopefully it is now clear that what is plotted is the mean accuracy of the LDA-decoding algorithm across permutations.

Edited Figure 2 legend:

"Figure 2: Ventrolateral prefrontal neurons maintain information for both spatial and reward stimuli during delay epochs. The mean performance of classifiers (1000 permutations, see Methods) trained to decode each task feature (spatial location, **a** and **c**; reward level, **b** and **d**) are plotted for each brain area and trial type (SR-task, **a** and **b**; RS-task, **c** and **d**). Ventrolateral prefrontal cortex (VLPFC) is the only region to strongly code information about space and reward across the trial. Notably, the VLPFC activity primarily encodes the factor most recently presented. When the reward cue is shown first (RS task, **c** and **d**), a representation of reward size is maintained throughout the first delay, but falls away when the spatial cue is presented. More surprisingly, a similar weakening of spatial coding is also observed on the SR Task (**a**), even though this analysis is restricted to trials where the subject remembered the correct spatial location. The VLPFC population strongly encodes and maintains a representation of the remembered spatial location, but this is substantially weakened by the offset of the reward cue. The first solid vertical line signifies when subjects were cued to respond. The first and second dashed vertical lines represent the average timing of the subjects' saccade and the onset of reward respectively. Solid coloured horizontal lines represent significant encoding for the corresponding brain region (2.5th percentile of distribution > chance level, $p < 0.05$). The dashed magenta line represents chance level classifier performance."

Added to Methods section:

"Decoding using Linear-discriminant analysis (LDA)

A decoder based upon LDA was used to predict task features(Aishwarya Parthasarathy et al., 2017). Decoding was performed separately for different task-types (i.e. SR or RS) and different task features (i.e. space and reward) in each neuronal population of interest. Neurons were pooled across sessions to create pseudopopulations. Neuronal firing rate was estimated at every 50ms using a 100ms window around the bin centre. This value was z-scored relative to the across-trial mean and standard deviation of firing rate in the final 300ms of the fixation period. Decoders were built to classify either spatial location or reward size. Chance level performance for the classifier was therefore 12.5% (8 spatial locations) and 20% (5 reward levels), respectively. Half of all correct trials, grouped with a uniform distribution across conditions and reorganised into 1500 pseudotrials, were used as a training set. In the majority of analyses, the remaining correct trials were used to construct

100 pseudotrials to be used as a test set. The one exception was the error-trial analysis (**Supplementary Fig.1**), where error trials were used to construct the test set. Both sets of data were denoised using principal components analysis (PCA) at each timepoint. The data were reassembled using the minimal number of components sufficient to explain 90% of the variance. The purpose of this pre-processing step was to avoid singular matrices when LDA was performed and to reduce noise when using the decoder. The data were then input to the Matlab function 'Classify'. This process was repeated 1000 times to create a distribution of classifier performance.

Initially, the relevant task condition (reward or space) was predicted by training and testing the decoder on datasets from equivalent timepoints (**Fig.2a-d; Fig.3**). Classifier performance at a particular timepoint was determined to be significant if the 2.5th percentile of the distribution exceeded chance (**Fig.2a-d**). To compare the classifier performance for high and low tau subpopulations (**Fig.3**), the classifier performance for each subpopulation was averaged across 500ms epoch windows within each permutation. The epochs were non-overlapping consecutive windows (0 to 500ms of Fixation; 500 to 1000ms of Fixation; 0 to 500ms of First Cue; 0 to 500ms of First Delay; 500-1000ms of First Delay; 0 to 500ms of Second Cue; 0 to 500ms of Second Delay; 500 to 1000ms of Second Delay; 0 to 500ms after go cue; 500ms to 1000ms after go cue). For each epoch, a bootstrap test compared the distribution between the two populations (Pagan, Urban, Wohl, & Rust, 2013). Specifically, the population with the highest average performance across all permutations was determined (typically the High Tau population in epochs of interest). We then calculated the number of permutations where the value for the population with the lower average (typically the Low Tau population) was above the value for the population with the higher average. The pairing of each permutation was arbitrary, so the pairings were randomly shuffled. This process was repeated 1000 times and the average was taken. The p-value was this number divided by the total number of permutations. This was then corrected for multiple comparisons (Bonferroni correction for 10 epochs). This same bootstrap test was used to compare classifier performance when tested on correct trials and error trials (**Supplementary Fig.1**).

To further probe the temporal evolution of the neural coding, we then extended our approach so that for each timepoint a decoder was trained at, this decoder was also tested at all other points within the trial. Averaging performance across permutations created a timepoints x timepoints matrix of classifier performance (**Supplementary Fig.2a**). To investigate the effect of the second stimulus on the neural code, we defined 3 time periods as follows; T1 (the final 500ms of Delay 1), Post Reward Cue On (250ms to 750ms after reward cue onset), and T2 (the final 500ms of Delay 2). The performance of a decoder trained in T1 and tested in T1 (T1T1), trained in T2 and tested in T2 (T2T2), trained in T1 and tested in T2 (T1T2), and trained in T2 and tested in T1 were compared (**Supplementary Fig.2e**). Performance was averaged within the 500ms time period for each permutation, then the difference between the two classifier performance distributions was compared to 0. If the 95th percentile range of the distribution did not overlap with 0, there was a significant difference in performance across time.

A final test to verify if the weakening of the spatial code in SR trials was caused by the presentation of the reward cue was employed (Aishwarya Parthasarathy et al., 2017). **Supplementary Fig. 2f** shows the average performance (across permutations) of all classifiers trained in T1 for all timepoints. For each of the three epoch windows (T1, Post

Reward Cue On, T2) a straight line was fitted to the data within each permutation. If the 95th percentile range of line gradients across permutations did not overlap with 0, there was a significant change in classifier performance (Supplementary Fig.2g)."

5. The finding of anti-correlation between cue and delay period activity was interesting, and the analysis in this context was novel. Such a pattern of activity has been described at the level of single neurons before (e.g. see reference 30).

We have taken a closer look at reference 30 of our initial submission (Qi et al. 2010, Frontiers in Systems Neuroscience). Whilst this study demonstrates that many single neurons in DLPFC exhibit strong cue-related responses that often diminish in the delay period (or only emerge in the delay period), as best we can tell, this paper does not discuss how the "selectivity pattern" changes between cue and delay. In our study, the way in which spatial (Figure 5) AND reward (Supplementary Fig 6) information is encoded reverses between cue and delay. We believe this is a novel result, and one that is different from the results shown in the Qi et al. study.

6. Line 258: "Demeaning" may be misconstrued for "belittling". Rephrase.

We have changed line 258 as follows:

*"By demeaning (subtracting the average; see **Methods**) activity across conditions for each neuron and repeating the analysis, we revealed an anticorrelation in the activity of high tau VLPFC neurons between the spatial cue and delay periods."*

The methods have also been extended to emphasise this point:

"Across-trial Correlation Analysis"

*Independent to selectivity measures, neural firing rate was correlated across the trial (**Fig.5a-b**). Firing rate for each condition (eight spatial locations, five reward levels) was correlated across neurons between each timepoint pair. A separate training and test set were defined based upon a split half of the trials. The matrix of correlation coefficients plotted represents the average (using Fisher's Z-transform) value across all of the conditions (**Fig.5a**). For **Fig.5b**, prior to performing the correlation, neural firing rate was demeaned within each condition at each timepoint for each neuron. Demeaning was performed at each timepoint, for each neuron; the mean firing rate across conditions was subtracted from the raw firing rate to generate the new value."*

The Figure 5 legend has also been clarified:

*"b) Within-condition correlation analysis where activity for each neuron was demeaned across each of the spatial locations (see **Methods**)."*

7. The authors grapple with the question of where the spatial information is actually maintained after the task-relevant distractor appears in the middle of the trial and disrupts stable encoding. They end the paper suggesting that a stable mnemonic representation may be transmitted to oculomotor regions and only maintained in the form of a motor plan. But would that not be the frontal eye field in the case of the task being studied?

We have now specifically referenced frontal eye field in our discussion as a downstream oculomotor region to which the spatial code could be transmitted.

“An important question therefore remains how WM is achieved on this task. The residual level of dynamic VLPFC spatial coding may be sufficient for WM performance. Another possibility, although not directly verifiable with our data, is that a PFC region maintains a representation of the mnemonic stimulus in an activity silent state (Mongillo et al., 2008; Stokes, 2015). Alternatively, PFC may be essential for setting up a stable mnemonic representation during the initial delay, but if this mnemonic information specifies a response that could be prepared, this information could then be transmitted to oculomotor regions, such as frontal eye field or superior colliculus, to prepare a saccade. This would be akin to activity in premotor regions for reaching movements (Cisek, 2007). It would be worthwhile to explore these ideas in a task where the contents of WM were independent of the response (Rainer & Miller, 2002; Romo et al., 1999; Warden & Miller, 2010), and investigate the impact of a salient distractor on the WM representation.”

8. Typo in line 538 (second).

This has been fixed.

References:

Buckley, M.J., Mansouri, F.A., Hoda, H., Mahboubi, M., Browning, P.G., Kwok, S.C., Phillips, A., and Tanaka, K. (2009). Dissociable components of rule-guided behavior depend on distinct medial and prefrontal regions. *Science* 325, 52-58.

Cavada, C., and Goldman-Rakic, P.S. (1989). Posterior parietal cortex in rhesus monkey: II. Evidence for segregated corticocortical networks linking sensory and limbic areas with the frontal lobe. *Journal of Comparative Neurology* 287, 422-445.

Reviewer #3 (Remarks to the Author):

The authors show how neurons in monkey PFC dynamically code spatial location and reward information. The description of previous literature is great, the results are generally well described as well, and the techniques and analysis methods seem sound. However, the results do not seem to contain a major new contribution, in particular relative to the recent papers by Jacob & Nieder 2014 and Parthasarathy et al. 2017 *Nature Neuroscience* (as also referenced in the manuscript).

We believe our results do represent a major new contribution. We hope the following explanation clarifies this.

Our results provide the most comprehensive analysis of the role of PFC in a task which allows different WM models (i.e. persistent activity vs. dynamic coding) to be directly compared. We recorded from four subregions of PFC in the same subjects, performing a task specifically designed to have an intervening (behaviourally relevant) stimulus. This has allowed our paper to present many novel results not explored in the above papers (Jacob and Nieder, 2014; Parthasarathy et al. 2017), or indeed elsewhere in the literature. Our results demonstrate:

- When directly comparing ACC, DLPFC, OFC and VLPFC on the same task – VLPFC has the strongest and most prolonged WM representations within PFC.
- We now show, through an error analysis, that VLPFC spatial selectivity is behaviourally relevant as it predicts correct performance.
- We demonstrate that a neuron's time constant (indexed by its spike-count autocorrelation at rest) is predictive of the dynamics by which it represents information during WM, but it does not predict a neuron's cue-related selectivity.
- However, we demonstrate that high tau cells *specifically in* VLPFC maintain information in WM in a stable code across delays.
- Our paper is the first to demonstrate that neurons within VLPFC invert their spatial and reward coding between cue presentation and the subsequent delay.
- We demonstrate that PFC WM dynamics are disrupted by a behaviourally relevant distractor, which challenges proposals that PFC attractors could support WM processes.
- Our results show, contrary to the Parthasarathy and Jacob papers and the majority of the WM literature, that a behaviourally relevant distractor not only disrupts WM representations – but it is actively encoded and maintained across time by PFC neurons during a WM task. This suggests that different neural mechanisms may be required to maintain WM when a distracting stimulus also carries behavioural relevance and activates neurons across PFC. This WM mechanism seemingly eludes current attractor models, which predict distractor-resistant spatial selectivity within PFC (Murray, Bernacchia, et al., 2017; Murray, Jaramillo, et al., 2017).
- Our paper is the best demonstration of how two disparate models of WM – persistent activity and dynamic coding – can be reconciled. We show strong evidence for persistent WM representations in VLPFC high tau neurons, but this WM activity only remains stable while the mnemonic stimulus is the location of the subject's attention. Once attention is redirected to other task-relevant information, the WM code is in a more dynamic format.
- We also show stimulus-locked, cross-task generalisation - but an absence of action-locked generalisation - of the spatial coding in VLPFC across the different trial types. These results indicate that a different set of read-out weights for WM of spatial location would be required from VLPFC activity for correct performance on the two trial types, implying multiple, independent task-specific neural states can support WM.
- Moreover, we provide one of the clearest demonstrations of how individual VLPFC neurons flexibly shift between encoding task-relevant representations.

Next, the reviewer highlighted in a later comment a further novel aspect about our study relative to the previous two papers:

There was one result that I found interesting which is not really emphasized in the manuscript, switching seems to happen mostly for RS trials (e.g. Figure 2d), not SR trials (e.g. Figure 2a). So reward coding is lost when location cue comes up, but not that clearly in the other direction. This seems to suggest that spatial information is encoded more robustly than reward information. At least in the vIPFC this seems a clear result. (This also points to an interesting difference relative to the two previous papers on the subject, the two cues do not carry the same type of information, so the second cue is not simply 'distracting' information, it is additional information that is also relevant for the animal.)

We agree that this is something we did not emphasise originally. We have added the following text to the results section:

“These results are consistent with VLPFC spiking-activity prioritising a representation of the most recently attended information, regardless of whether it is necessary to store the stimulus in WM for successful performance(Lebedev et al., 2004; Watanabe & Funahashi, 2014). However, while the spatial coding is weakened by reward cue presentation on SR trials (Fig.2a), reward coding is reduced to insignificance by spatial cue presentation on RS trials (Fig.2d). This suggests that following the presentation of a subsequent stimulus, VLPFC maintains a residual level of coding for relevant, but not irrelevant, information.”

Furthermore, we believe our paper significantly builds upon some important unresolved points raised in the Jacob and Parthasarathy papers. From their studies, it is unclear in what context PFC neurons encode distractors. Our paper is the first to demonstrate that PFC neurons encode distractor information while a spatial location is held in working memory.

The Jacob paper shows that neurons within PFC preferentially encode distractor information at the expense of working memory information. The factor to maintain within working memory is a numerosity level, and the distracting stimulus was a subsequently presented numerosity cue. Importantly, the distracting stimulus is centrally presented and would be task relevant if presented earlier in the trial. The Parthasarathy paper shows that neurons within PFC do not encode distractor information at the expense of working memory information; although the working memory code is morphed as a consequence. The factor to maintain in working memory is a spatial saccade target, and the distracting stimulus was also a saccade target. Importantly, the distractor saccade target is both peripherally presented and a different colour to the initial saccade target. Therefore, unlike the Jacob paper, the distractor cue could never act as the memory location – and committing that stimulus to working memory is never beneficial.

The discrepancy between these papers, of whether PFC encodes distractors, could therefore be due to a difference in the information required to be held in working memory (spatial location vs. numerosity), or if the cue is centrally presented, or indeed whether that cue would ever carry task-useful information.

	Item held in WM	Location of Distractor	Distractor could carry behavioural relevance if presented earlier in the trial
Jacob and Nieder	Numerosity	Central	Yes
Parthasarathy	Saccade target	Peripheral	No

The paper by Parthasarathy acknowledges this discrepancy. They say:

‘A previous study found a sharp decrease in target information following distractor presentation, together with an increase of distractor information (Jacob & Nieder, 2014). Our results, however, did not replicate these observations. Rather, we found that target information remained stable, and distractor information stayed close to baseline throughout the trial (Supplementary Fig. 1c). This difference may reflect the simpler nature of our task and the comparatively lower behavioral saliency of the distractor we used. It may also reflect differences in the ways different types of information are encoded; perhaps the working memory code for numerosity in LPFC is more susceptible to distractors than the code for spatial locations’.

These task differences therefore leave an unresolved question of when PFC is resistant to encoding distractor information (as referred to in Parthasarathy et al. 2017). It could be that PFC does not encode distractor information during a spatial working memory task, or PFC does not encode information which is entirely task-irrelevant. From the literature alone, the evidence suggests the former would be the more likely assumption, as multiple studies have shown no PFC distractor-encoding during spatial working memory tasks (di Pellegrino & Wise, 1993; Aishwarya Parthasarathy et al., 2017; Qi et al., 2010; Suzuki & Gottlieb, 2013). However, our results resolve an important discrepancy in the literature, as we are the first to demonstrate that PFC neurons encode distractor information while a spatial location is held in working memory.

In summary, we believe our study provides a number of novel findings, and makes an important contribution to the WM literature.

The paper claims that there is a switching between the coding for spatial location and the reward, but this seems more like a general decline, in particular for SR trials. There is no comparison with trials without a second cue (the reward cue in SR trials and the location cue in the RS trials), which makes it hard to directly compare the influence of this second cue. Furthermore, no effort is made to quantify how abrupt the change was (as for example in Figure 2b and f in Parthasarathy et al. 2017 Nature Neuroscience).

The reviewer raises an important point here that we didn’t mention in the original manuscript. We agree that including trials without a second cue would allow us to directly compare the influence of a distracting stimulus with the normal temporal evolution of the working memory code. Unfortunately, the experiment did not contain this potentially stronger control condition, and hence trials without a second cue were not included.

However, as in the paper by Parthasarathy and colleagues, we have now employed an alternative method to quantify the effect of the distractor on the stability of the working memory code. We thank the reviewer for suggesting this additional analysis and believe we now present strong evidence that there is a distractor-triggered weakening of the spatial code on SR trials (see below).

We have amended the text in the results section as follows:

*“To confirm whether the ‘disruptive’ influence of the reward cue was quantitatively different from the normal temporal evolution of the WM code, it would have been ideal to include a condition where the spatial information had to be remembered for the same length of time without an interceding reward cue. Unfortunately, this condition was not included in the experimental design, so an alternative approach was used (**Supplementary Fig.2**)(Aishwarya Parthasarathy et al., 2017). This analysis demonstrated that spatial coding was significantly reduced in delay 2 relative to delay 1 (**Supplementary Fig.2C-E**), and that this decline in stability was triggered by the presentation of the reward cue (**Supplementary Fig.2F-G**).”*

Supplementary Figure 2: Quantifying temporal evolution of VLPFC spatial code on SR Trials. **a)** Cross-temporal decoding performance of spatial coding (see **Methods**). Annotated squares show time points used to compare decoding performance across epochs (T1T1 - Decoder trained in delay 1, tested in delay 1; T2T2 - Decoder trained in delay 2, tested in delay 2; T1T2 - Decoder trained in delay 1, tested in delay 2; T2T1 - Decoder trained in delay 2, tested in delay 1.) **b)** Heatmap plotting the change in cross-temporal decoding performance between timepoint (t) and a bin three timepoints later. The annotated square shows T1T1, with the lines extending from it representing the onset of the reward cue. Reward cue onset seems to reduce decoding performance. **c)** Across-trial performance of all classifiers trained within the T1 window. **d)** Across-trial performance of all classifiers trained within the T2 window. **e)** Boxplot comparing the performance of different classifiers across 1000 permutations. T1T1-T2T2 (first bar) shows that spatial coding is significantly reduced from delay one to delay two. (***, $p < 0.001$; *, $p < 0.05$) **f)** Average performance of all classifiers trained within the T1 window. Black boxes show the gradient of the performance across time. Spatial coding is fairly stable by the end of delay 1, but a sharp drop follows the onset of the reward cue. **g)** Boxplots show the distributions of these gradients across permutations. The slope fitted following the reward cue onset is significantly negative (***, $p = 0.001$), showing a decrease in spatial coding. The other slopes were not significantly different from 0. The area contained within the whiskers of the boxplots represents the 95th percentile range of the distributions. The central mark is the median of the distribution.

The analysis of low versus high tau are novel, but the results are not that surprising. There is just a graded effect, neurons with long time constants show longer sustained activity than neurons with short time constants. As also mentioned in the manuscript, "it is not the case that low tau subpopulations are simply less task-selective" [line 186].

We appreciate the reviewer's point, but we hope to clarify a couple of ideas.

First, our comment on line 186 was intended to emphasise that it wasn't simply the case that tau correlated with task-related selectivity; indeed, there was no significant difference in the high and low tau subpopulations for encoding cue-related information (Fig. 3). But in looking at the temporal profile of this selectivity pattern, one will notice that arguably both the low and high tau populations have "graded" responses at the cue, peaking at around 200-ms post-cue (Fig. 3). Thus, tau does not predict cue-related selectivity. In contrast, what clearly separates high and low tau subpopulations in this dataset is that only high tau cells maintained the cue information in a stable code during the initial working memory delay (see Figure 4). Moreover, it wasn't the case that high tau predicts working memory selectivity in all neurons, as neurons in OFC and ACC did not encode information during working memory delays (more on this below). Thus, our results demonstrate that within VLPFC, a neuron's time constant is predictive of the dynamics by which it represents information during working memory, but does not predict a neuron's cue-related selectivity.

Second, as the time constant is characterised based upon a neuron's resting activity (during the fixation period), there is no a priori reason to assume this will relate to its task-related encoding of working memory information; indeed the time constant fitting is done blind to neuron identity or functional properties. The fact that neurons which do have a prolonged autocorrelation are more involved in supporting working memory reveals important information of the circuit-level mechanisms underlying working memory.

Furthermore, it is clear that whether a neuron displays sustained working memory activity is not just dependent upon its time constant. In ACC, the brain area with the longest taus, there was no prolonged maintenance of spatial information (Fig2). This suggests a combination of a neuron's tau, and the function of the brain region, determines if it will have prolonged working memory correlates.

We have added to our discussion as follows:

"However, ACC, the PFC region where the longest taus were observed, did not display prolonged selectivity for either reward or spatial information. This may suggest that there is an optimal tau required to support WM. However, an alternate explanation is that a neuron's selectivity pattern is constrained by both the functional anatomy of its brain region and its tau. Long tau neurons in ACC may perform complex functions across longer timescales than our task was designed to investigate, such as the integration of information across trials(Bernacchia et al., 2011)."

The comparison between areas is potentially interesting, but is not really exploited. The main effect that is found is that it's mostly the vLPFC that codes for spatial location. This is rather surprising with respect to previous literature which indicates that the dLPFC has a preference for spatial location while the vLPFC has a preference for object identity (e.g. Wilson, Scialdhe &

Goldman-Rakic, Science 1993; Riley, Qi & Constantinidis Cerebral Cortex 2016).

We agree with the reviewer that there may be some surprise that vLPFC has much stronger spatial tuning than in dLPFC in our study. We have discussed this point extensively in our response to reviewer 2 (see above), but we have adapted our discussion point to place this finding in the context of the wider literature:

“Of the PFC regions studied, VLPFC mnemonic representations were the strongest, and the only ones present during the second delay of SR trials, although in an altered state relative to the initial delay. DLPFC had relatively weaker and short-lived coding of spatial location (although more prolonged than ACC or OFC). This is perhaps surprising, as some hypotheses propose DLPFC primarily maintains WM for spatial locations, whereas VLPFC maintains WM for object identities (Riley et al., 2016; Wilson et al., 1993). However, strong connections exist between parietal cortex and VLPFC (Cavada & Goldman-Rakic, 1989; Petrides & Pandya, 1984) which could provide spatial information, and VLPFC receives input from high-level visual areas in the temporal lobe (Barbas, 1988; Petrides & Pandya, 1999) which could provide object-related information. Perhaps consequently, other reports, including our previous single neuron analyses of this dataset (Kennerley & Wallis, 2009b), observe equivalent or stronger spatial selectivity ventral to the principal sulcus (Hoshi et al., 2000; Lebedev et al., 2004; Rao et al., 1997). These studies all required subjects to attend to spatial information in addition to other task features (e.g., reward information, object identity, task rule). Thus, it is possible that spatial selectivity may shift toward VLPFC in contexts where flexible allocation of attention to multiple task features is required. Indeed, the flexible shifting of VLPFC selectivity that we observed (Fig. 7) is consistent with the idea that VLPFC encodes the focus of attention (Lebedev et al., 2004).”

There was one result that I found interesting which is not really emphasized in the manuscript, switching seems to happen mostly for RS trials (e.g. Figure 2d), not SR trials (e.g. Figure 2a). So reward coding is lost when location cue comes up, but not that clearly in the other direction. This seems to suggest that spatial information is encoded more robustly than reward information. At least in the vLPFC this seems a clear result. (This also points to an interesting difference relative to the two previous papers on the subject, the two cues do not carry the same type of information, so the second cue is not simply 'distracting' information, it is additional information that is also relevant for the animal.)

We are grateful for the reviewer for raising this point. We have modified the results section as follows:

“These results are consistent with VLPFC spiking-activity prioritising a representation of the most recently attended information, regardless of whether it is necessary to store the stimulus in WM for successful performance (Lebedev et al., 2004; Watanabe & Funahashi, 2014). However, while the spatial coding is weakened by reward cue presentation on SR trials (Fig. 2a), reward coding is reduced to insignificance by spatial cue presentation on RS trials (Fig. 2d). This suggests that following the presentation of a subsequent stimulus, VLPFC maintains a residual level of coding for relevant, but not irrelevant, information.”

This makes it potentially interesting to look more closely at ACC, as this area codes more robustly for the reward cue than the location cue. Unfortunately, the coding for location is so weak that it is hard to judge whether the switch happens in the SR trials in that case. At least for the ACC there seems to be no clear switch for RS trials (Figure 2d).

The reviewer raises an interesting point; the weakening of reward coding on RS trials in ACC appears much more like a gradual decline, as opposed to a decrease triggered by the spatial cue. However, spatial coding in ACC is very weak, so it would be difficult to qualitatively or quantitatively compare the switching between the two trials.

This comment did give us the idea to compare the decline in reward coding on RS trials (Fig2D) between VLPFC and ACC. In VLPFC, there is a clear drop in reward information timelocked to the onset of the spatial cue (**Response to Reviewers Fig. 1**). In ACC, as the reviewer suggested, there seems to be no clear switch; rather an ongoing temporal decline (**Response to Reviewers Fig. 2**).

Response to Reviewers Figure 1: Quantifying temporal evolution of VLPFC reward code on RS Trials. A) Cross-temporal decoding performance of reward coding (see **Methods**). Annotated squares show time points used to compare decoding performance across epochs (T1T1 - Decoder trained in delay 1, tested in delay 1; T2T2 - Decoder trained in delay 2, tested in delay 2; T1T2 - Decoder trained in delay 1, tested in delay 2; T2T1 - Decoder trained in delay 2, tested in delay 1.) B) Heatmap plotting the change in cross-temporal decoding performance between timepoint (t) and a bin three timepoints later. The annotated square shows T1T1, with the lines extending from it the onset of the spatial cue. This seems to reduce decoding performance. C) Cross-trial performance of all classifiers trained within the T1 window. The black bars show the 95% confidence interval for classifier. D) Cross-trial performance of all classifiers trained within the T2 window. E) Boxplot comparing the performance of different classifiers across 1000 permutations. T1T1-T2T2 (first bar) shows that reward

coding is significantly reduced from delay one to delay two. F) Average performance of all classifiers trained within the T1 window. Black boxes show the gradient of the performance across time. Reward coding is fairly stable by the end of delay 1, but a sharp drop follows the onset on the spatial cue. G) Boxplots show the distributions of these gradients across permutations. The slope fitted following the spatial cue onset is significantly negative, showing a decrease in reward coding. The other slopes were not significantly different from 0.

Response to Reviewers Figure 2: Quantifying temporal evolution of ACC reward code on RS Trials. A) Cross-temporal decoding performance of reward coding (see **Methods**). Annotated squares show time points used to compare decoding performance across epochs (T1T1 - Decoder trained in delay 1, tested in delay 1; T2T2 - Decoder trained in delay 2, tested in delay 2; T1T2 - Decoder trained in delay 1, tested in delay 2; T2T1 - Decoder trained in delay 2, tested in delay 1.) B) Heatmap plotting the change in cross-temporal decoding performance between timepoint (t) and a bin three timepoints later. The annotated square shows T1T1, with the lines extending from it the onset of the spatial cue. This does not seem to dramatically reduce decoding performance. C) Cross-trial performance of all classifiers trained within the T1 window. The black bars show the 95% confidence interval for classifier. D) Cross-trial performance of all classifiers trained within the T2 window. E) Boxplot comparing the performance of different classifiers across 1000 permutations. T1T1-T2T2 (first bar) shows that reward coding is significantly reduced from delay one to delay two. F) Average performance of all classifiers trained within the T1 window. Black boxes show the gradient of the performance across time. Reward coding is

gradually declining throughout the trial, there is no sharp drop following the onset on the spatial cue. G) Boxplots show the distributions of these gradients across permutations. The slope fitted following the spatial cue onset is not significantly different from 0.

minor comment:

It might be helpful to represent cross-temporal plots with a white background, see

https://github.com/kingjr/fix_your_jet

Thank you to the reviewer for this suggestion. We have recreated Figure 4, Figure 6, and the previous Supplementary Figure 1 and Supplementary Figure 2 on a white background. We decided to maintain the current colour scheme for Figure 5 and previous Supplementary Figure 3. This was because we wanted to clearly demonstrate that the initial within-condition correlation of neural firing (Fig5A and former Supplementary Figure 3.A) is positive for all timepoints; whereas the demeaned version is negative when cue activity is projected onto the first delay (asterisks in Fig5B and former Supplementary Figure 3.B). If both plots are displayed on a white background, we believe this subtle difference would not come across clearly.

- Barbas, H. (1988). Anatomic organization of basoventral and mediodorsal visual recipient prefrontal regions in the rhesus monkey. *J Comp Neurol*, 276(3), 313-342. doi:10.1002/cne.902760302
- Bauer, R. H., & Fuster, J. M. (1976). Delayed-matching and delayed-response deficit from cooling dorsolateral prefrontal cortex in monkeys. *J Comp Physiol Psychol*, 90(3), 293-302.
- Bernacchia, A., Seo, H., Lee, D., & Wang, X. J. (2011). A reservoir of time constants for memory traces in cortical neurons. *Nat Neurosci*, 14(3), 366-372. doi:10.1038/nn.2752
- Blackman, R. K., Crowe, D. A., DeNicola, A. L., Sakellaridi, S., MacDonald, A. W., 3rd, & Chafee, M. V. (2016). Monkey Prefrontal Neurons Reflect Logical Operations for Cognitive Control in a Variant of the AX Continuous Performance Task (AX-CPT). *J Neurosci*, 36(14), 4067-4079. doi:10.1523/JNEUROSCI.3578-15.2016
- Brunel, N., & Wang, X. J. (2001). Effects of neuromodulation in a cortical network model of object working memory dominated by recurrent inhibition. *J Comput Neurosci*, 11(1), 63-85.
- Cavada, C., & Goldman-Rakic, P. S. (1989). Posterior parietal cortex in rhesus monkey: II. Evidence for segregated corticocortical networks linking sensory and limbic areas with the frontal lobe. *J Comp Neurol*, 287(4), 422-445. doi:10.1002/cne.902870403
- Cavanagh, S. E., Wallis, J. D., Kennerley, S. W., & Hunt, L. T. (2016). Autocorrelation structure at rest predicts value correlates of single neurons during reward-guided choice. *Elife*, 5. doi:10.7554/eLife.18937
- Cisek, P. (2007). Cortical mechanisms of action selection: the affordance competition hypothesis. *Philos Trans R Soc Lond B Biol Sci*, 362(1485), 1585-1599. doi:10.1098/rstb.2007.2054
- di Pellegrino, G., & Wise, S. P. (1993). Visuospatial versus visuomotor activity in the premotor and prefrontal cortex of a primate. *J Neurosci*, 13(3), 1227-1243.
- Enel, P., Procyk, E., Quilodran, R., & Dominey, P. F. (2016). Reservoir Computing Properties of Neural Dynamics in Prefrontal Cortex. *PLoS Comput Biol*, 12(6), e1004967. doi:10.1371/journal.pcbi.1004967
- Freedman, D. J., Riesenhuber, M., Poggio, T., & Miller, E. K. (2003). A comparison of primate prefrontal and inferior temporal cortices during visual categorization. *J Neurosci*, 23(12), 5235-5246.

- Funahashi, S., Bruce, C. J., & Goldman-Rakic, P. S. (1989). Mnemonic coding of visual space in the monkey's dorsolateral prefrontal cortex. *J Neurophysiol*, *61*(2), 331-349. doi:10.1152/jn.1989.61.2.331
- Funahashi, S., Bruce, C. J., & Goldman-Rakic, P. S. (1990). Visuospatial coding in primate prefrontal neurons revealed by oculomotor paradigms. *J Neurophysiol*, *63*(4), 814-831. doi:10.1152/jn.1990.63.4.814
- Funahashi, S., Bruce, C. J., & Goldman-Rakic, P. S. (1991). Neuronal activity related to saccadic eye movements in the monkey's dorsolateral prefrontal cortex. *J Neurophysiol*, *65*(6), 1464-1483. doi:10.1152/jn.1991.65.6.1464
- Funahashi, S., Bruce, C. J., & Goldman-Rakic, P. S. (1993). Dorsolateral prefrontal lesions and oculomotor delayed-response performance: evidence for mnemonic "scotomas". *J Neurosci*, *13*(4), 1479-1497.
- Hoshi, E., Shima, K., & Tanji, J. (2000). Neuronal activity in the primate prefrontal cortex in the process of motor selection based on two behavioral rules. *J Neurophysiol*, *83*(4), 2355-2373. doi:10.1152/jn.2000.83.4.2355
- Jacob, S. N., & Nieder, A. (2014). Complementary roles for primate frontal and parietal cortex in guarding working memory from distractor stimuli. *Neuron*, *83*(1), 226-237. doi:10.1016/j.neuron.2014.05.009
- Kennerley, S. W., & Wallis, J. D. (2009a). Encoding of reward and space during a working memory task in the orbitofrontal cortex and anterior cingulate sulcus. *J Neurophysiol*, *102*(6), 3352-3364. doi:10.1152/jn.00273.2009
- Kennerley, S. W., & Wallis, J. D. (2009b). Reward-dependent modulation of working memory in lateral prefrontal cortex. *J Neurosci*, *29*(10), 3259-3270. doi:10.1523/JNEUROSCI.5353-08.2009
- Lebedev, M. A., Messinger, A., Kralik, J. D., & Wise, S. P. (2004). Representation of attended versus remembered locations in prefrontal cortex. *PLoS Biol*, *2*(11), e365. doi:10.1371/journal.pbio.0020365
- Lennert, T., & Martinez-Trujillo, J. (2011). Strength of response suppression to distracter stimuli determines attentional-filtering performance in primate prefrontal neurons. *Neuron*, *70*(1), 141-152. doi:10.1016/j.neuron.2011.02.041
- Lundqvist, M., Rose, J., Herman, P., Brincat, S. L., Buschman, T. J., & Miller, E. K. (2016). Gamma and Beta Bursts Underlie Working Memory. *Neuron*, *90*(1), 152-164. doi:10.1016/j.neuron.2016.02.028
- Meder, D., Kolling, N., Verhagen, L., Wittmann, M. K., Scholl, J., Madsen, K. H., . . . Rushworth, M. F. S. (2017). Simultaneous representation of a spectrum of dynamically changing value estimates during decision making. *Nat Commun*, *8*(1), 1942. doi:10.1038/s41467-017-02169-w
- Mendoza-Halliday, D., & Martinez-Trujillo, J. C. (2017). Neuronal population coding of perceived and memorized visual features in the lateral prefrontal cortex. *Nat Commun*, *8*, 15471. doi:10.1038/ncomms15471
- Meyers, E. M., Freedman, D. J., Kreiman, G., Miller, E. K., & Poggio, T. (2008). Dynamic population coding of category information in inferior temporal and prefrontal cortex. *J Neurophysiol*, *100*(3), 1407-1419. doi:10.1152/jn.90248.2008
- Mongillo, G., Barak, O., & Tsodyks, M. (2008). Synaptic theory of working memory. *Science*, *319*(5869), 1543-1546. doi:10.1126/science.1150769
- Murray, J. D., Bernacchia, A., Roy, N. A., Constantinidis, C., Romo, R., & Wang, X. J. (2017). Stable population coding for working memory coexists with heterogeneous neural dynamics in prefrontal cortex. *Proceedings of the National Academy of Sciences of the United States of America*, *114*(2), 394-399. doi:10.1073/pnas.1619449114

- Murray, J. D., Jaramillo, J., & Wang, X. J. (2017). Working Memory and Decision-Making in a Frontoparietal Circuit Model. *J Neurosci*, *37*(50), 12167-12186. doi:10.1523/JNEUROSCI.0343-17.2017
- Pagan, M., Urban, L. S., Wohl, M. P., & Rust, N. C. (2013). Signals in inferotemporal and perirhinal cortex suggest an untangling of visual target information. *Nat Neurosci*, *16*(8), 1132-1139. doi:10.1038/nn.3433
- Parthasarathy, A., Herikstad, R., Bong, J. H., Medina, F. S., Libedinsky, C., & Yen, S.-C. (2017). Mixed selectivity morphs population codes in prefrontal cortex. *Nature Neuroscience*, *20*(12), 1770-1779. doi:10.1038/s41593-017-0003-2
- Parthasarathy, A., Herikstad, R., Bong, J. H., Medina, F. S., Libedinsky, C., & Yen, S. C. (2017). Mixed selectivity morphs population codes in prefrontal cortex. *Nat Neurosci*, *20*(12), 1770-1779. doi:10.1038/s41593-017-0003-2
- Paxinos, G., Huang, X. F., & Toga, A. W. (2000). *The Rhesus Monkey Brain in Stereotaxic Coordinates*: Academic Press.
- Petrides, M., & Pandya, D. N. (1984). Projections to the frontal cortex from the posterior parietal region in the rhesus monkey. *J Comp Neurol*, *228*(1), 105-116. doi:10.1002/cne.902280110
- Petrides, M., & Pandya, D. N. (1999). Dorsolateral prefrontal cortex: comparative cytoarchitectonic analysis in the human and the macaque brain and corticocortical connection patterns. *Eur J Neurosci*, *11*(3), 1011-1036.
- Qi, X. L., Katsuki, F., Meyer, T., Rawley, J. B., Zhou, X., Douglas, K. L., & Constantinidis, C. (2010). Comparison of neural activity related to working memory in primate dorsolateral prefrontal and posterior parietal cortex. *Front Syst Neurosci*, *4*, 12. doi:10.3389/fnsys.2010.00012
- Rainer, G., Asaad, W. F., & Miller, E. K. (1998). Memory fields of neurons in the primate prefrontal cortex. *Proceedings of the National Academy of Sciences of the United States of America*, *95*(25), 15008-15013.
- Rainer, G., & Miller, E. K. (2002). Timecourse of object-related neural activity in the primate prefrontal cortex during a short-term memory task. *Eur J Neurosci*, *15*(7), 1244-1254.
- Rao, S. C., Rainer, G., & Miller, E. K. (1997). Integration of what and where in the primate prefrontal cortex. *Science*, *276*(5313), 821-824.
- Raposo, D., Kaufman, M. T., & Churchland, A. K. (2014). A category-free neural population supports evolving demands during decision-making. *Nat Neurosci*, *17*(12), 1784-1792. doi:10.1038/nn.3865
- Rigotti, M., Barak, O., Warden, M. R., Wang, X. J., Daw, N. D., Miller, E. K., & Fusi, S. (2013). The importance of mixed selectivity in complex cognitive tasks. *Nature*, *497*(7451), 585-590. doi:10.1038/nature12160
- Riley, M. R., Qi, X. L., & Constantinidis, C. (2016). Functional specialization of areas along the anterior-posterior axis of the primate prefrontal cortex. *Cereb Cortex*. doi:10.1093/cercor/bhw190
- Romo, R., Brody, C. D., Hernandez, A., & Lemus, L. (1999). Neuronal correlates of parametric working memory in the prefrontal cortex. *Nature*, *399*(6735), 470-473. doi:10.1038/20939
- Saleem, K. S., Miller, B., & Price, J. L. (2014). Subdivisions and connectional networks of the lateral prefrontal cortex in the macaque monkey. *J Comp Neurol*, *522*(7), 1641-1690. doi:10.1002/cne.23498
- Spaak, E., Watanabe, K., Funahashi, S., & Stokes, M. G. (2017). Stable and Dynamic Coding for Working Memory in Primate Prefrontal Cortex. *J Neurosci*, *37*(27), 6503-6516. doi:10.1523/JNEUROSCI.3364-16.2017
- Stokes, M. G. (2015). 'Activity-silent' working memory in prefrontal cortex: a dynamic coding framework. *Trends Cogn Sci*, *19*(7), 394-405. doi:10.1016/j.tics.2015.05.004
- Stokes, M. G., Kusunoki, M., Sigala, N., Nili, H., Gaffan, D., & Duncan, J. (2013). Dynamic coding for cognitive control in prefrontal cortex. *Neuron*, *78*(2), 364-375. doi:10.1016/j.neuron.2013.01.039

- Suzuki, M., & Gottlieb, J. (2013). Distinct neural mechanisms of distractor suppression in the frontal and parietal lobe. *Nat Neurosci*, *16*(1), 98-104. doi:10.1038/nn.3282
- Wang, X. J. (1999). Synaptic basis of cortical persistent activity: the importance of NMDA receptors to working memory. *J Neurosci*, *19*(21), 9587-9603.
- Warden, M. R., & Miller, E. K. (2010). Task-dependent changes in short-term memory in the prefrontal cortex. *J Neurosci*, *30*(47), 15801-15810. doi:10.1523/JNEUROSCI.1569-10.2010
- Watanabe, K., & Funahashi, S. (2014). Neural mechanisms of dual-task interference and cognitive capacity limitation in the prefrontal cortex. *Nat Neurosci*, *17*(4), 601-611. doi:10.1038/nn.3667
- Wilson, F. A., Scaldie, S. P., & Goldman-Rakic, P. S. (1993). Dissociation of object and spatial processing domains in primate prefrontal cortex. *Science*, *260*(5116), 1955-1958.

Reviewers' Comments:

Reviewer #1 (Remarks to the Author):

The authors have addressed all my comments satisfactorily. I am happy to support publication of the manuscript in its present form.

Reviewer #2 (Remarks to the Author):

Authors have addressed my criticism.

Reviewer #3 (Remarks to the Author):

The authors convincingly indicated the relevance of their work, in particular by comparing it in more detail to previous studies. I agree that there are multiple aspects of the work that are novel, which together make it an important addition to the field.

minor comments:

- Why remove the 'RS task'/'SR task' labels above the panels in Figures 2 and 3, and the labels 'High Tau'/'Low Tau' in Figure 4? I thought it was better as it was. Same for Figure S3 and S5.
- Please add whether which axis represent the test window and which the training window in all cross-temporal plots.

REVIEWERS' COMMENTS:

Reviewer #1 (Remarks to the Author):

The authors have addressed all my comments satisfactorily. I am happy to support publication of the manuscript in its present form.

We are pleased the reviewer is happy to support the publication of our work. The reviewer provided constructive comments which improved the manuscript.

Reviewer #2 (Remarks to the Author):

Authors have addressed my criticism.

We are pleased the reviewer believes we addressed their criticism.

Reviewer #3 (Remarks to the Author):

The authors convincingly indicated the relevance of their work, in particular by comparing it in more detail to previous studies. I agree that there are multiple aspects of the work that are novel, which together make it an important addition to the field.

We are pleased the reviewer acknowledges the important novel contributions of the manuscript.

minor comments:

- Why remove the 'RS task'/'SR task' labels above the panels in Figures 2 and 3, and the labels 'High Tau'/'Low Tau' in Figure 4? I thought it was better as it was. Same for Figure S3 and S5.

Thank you for this point, we agree this would make the figures clearer. This point has now been addressed.

- Please add whether which axis represent the test window and which the training window in all cross-temporal plots.

These labels have been added for all cross-temporal plots. This includes the following figures:

- Figure4B-E
- Figure5A-B

- Figure6B (was previously included for this figure)
- Supplementary Figure 2A-B
- Supplementary Figure 4A-H
- Supplementary Figure 5A-B
- Supplementary Figure 6A-B